# Sulphostin-inspired *N*-phosphonopiperidones as selective covalent DPP8 and DPP9 inhibitors

Leonard Sewald [1,10], Werner W. A. Tabak [1,10], Lorenz Fehr [1,10], Samuel Zolg[2], Maja Najdzion[1], Carlo J. A. Verhoef [1,8], David Podlesainski [1,9], Ruth Geiss-Friedlander [2], Alfred Lammens[3], Farnusch Kaschani [1], Doris Hellerschmied [4], Robert Huber[5,6,7] & Markus Kaiser [1]✉

Covalent chemical probes and drugs combine unique pharmacologic properties with the availability of straightforward compound profiling technologies via chemoproteomic platforms. These advantages have fostered the development of suitable electrophilic "warheads" for systematic covalent chemical probe discovery. Despite undisputable advances in the last years, the targeted development of proteome-wide selective covalent probes remains a challenge for dipeptidyl peptidase (DPP) 8 and 9 (DPP8/9), intracellular serine hydrolases of the pharmacologically relevant dipeptidyl peptidase 4 activity/structure homologues (DASH) family. Here, we show the exploration of the natural product Sulphostin, a DPP4 inhibitor, as a starting point for DPP8/9 inhibitor development. The generation of Sulphostin-inspired *N*-phosphonopiperidones leads to derivatives with improved DPP8/9 inhibitory potency, an enhanced proteome-wide selectivity and confirmed DPP8/9 engagement in cells, thereby representing that structural fine-tuning of the warhead's leaving group may represent a straightforward strategy for achieving target selectivity in exoproteases such as DPPs.

Covalent chemical probes and drugs have experienced an impressive renaissance in the last years[1]. Their re-interest is based on their unique and distinct pharmacological properties, in combination with the availability of streamlined efficient chemoproteomic platforms for quantifying target engagement(s) or proteome-wide selectivities[2,3]. Consequently, a suite of electrophilic biocompatible "warheads", these are the chemical moieties with fine-tuned chemical reactivities to limit covalent bond formation to the small molecule binding hot spots of a protein, have been developed for tailored-targeting of active site residues and other hyperreactive residues on proteins[4]. Despite these advances, the development of proteome-wide selective covalent small molecules, in particular for proteins with homologues binding sites, has however remained a challenge.

This is particular true for serine hydrolases, a large family of enzymes characterized by the presence of an active site serine and featuring many clinically established or currently investigated drug

[1]Chemical Biology, Center of Medical Biotechnology, Faculty of Biology, University Duisburg-Essen, Essen, Germany. [2]Institute of Molecular Medicine and Cell Research, Faculty of Medicine, University of Freiburg, Freiburg, Germany. [3]Proteros Biostructures GmbH, Martinsried, Germany. [4]Mechanistic Cell Biology, Center of Medical Biotechnology, Faculty of Biology, University Duisburg-Essen, Essen, Germany. [5]Center of Medical Biotechnology, Faculty of Biology, University Duisburg-Essen, Essen, Germany. [6]Max-Planck-Institute of Biochemistry, Martinsried, Germany. [7]TUM Senior Excellence Faculty, Technical University of Munich, Munich, Germany. [8]Present address: Laboratory of Chemical Biology, Department of Biomedical Engineering, Institute for Complex Molecular Systems, Eindhoven University of Technology, MB Eindhoven, The Netherlands. [9]Present address: Faculty of Biology and Biotechnology, Ruhr University Bochum, Bochum, Germany. [10]These authors contributed equally: Leonard Sewald, Werner W. A. Tabak, Lorenz Fehr. ✉e-mail: markus.kaiser@uni-due.de

targets[5]. Due to its pharmacological relevance, many covalent chemical probes for serine hydrolases have already been developed. Most of them are based on the phosphonate moiety, the "gold standard" warhead for serine hydrolases, as phosphonates show unique covalent labeling properties in combination with a high pan-selectivity for serine hydrolases[6]. Their further development into a target-selective probe is however much more challenging and requires structural fine-tunings similar to a medicinal chemistry compound optimization campaign[7].

The dipeptidyl peptidases (DPP) 8 and 9 are serine proteases of the dipeptidyl peptidase 4 activity/structure homologues (DASH) subfamily which cleave post-proline bonds two residues from the N-terminus of a substrate[8–10]. Clinically established drug targets of this subfamily are DPP4 that is targeted by diverse anti-diabetes drugs due to its role in incretin proteolysis[11,12] and the Fibroblast activation protein-α (FAP) that is a target for tumor-specific delivery of antibody-conjugated drugs (ACDs)[13]. While DPP4 and FAP are extracellular or secreted proteases, DPP8 and DPP9 are located intracellularly to the cytoplasm and nucleus. There, they play important roles in diverse biological processes such as pyroptosis or ubiquitin-mediated proteolysis, turning them into promising targets for chemical probe and anti-inflammatory or anticancer drug discovery[14–16]. They are highly homologous, dimeric proteins (58% overall identity and 92% active site identity), and are built up from an N-terminal β-propeller domain carrying most of the ligand-binding residues and a C-terminal α/β hydrolase domain containing the catalytic triad composed of a serine, aspartate and histidine residue[17,18]. They have partially overlapping but also diverse physiological functions, which are however not yet fully understood. Among the two enzymes, DPP9 has been studied more extensively, particularly for its key role in inflammatory cell death, i.e., pyroptosis induction[19–23], or, as we have recently shown, for promoting the repair of DNA double strand breaks (DSBs)[24]. The active sites of DPP8 and DPP9 also share strong structural similarities to the one of DPP4 (Fig. 1a). However, DPP4's S2 substrate binding site (Schechter and Berger nomenclature[25]) is slightly smaller than the S2 site of DPP8/9, enabling to achieve DPP8/9 selectivity vs. DPP4 by incorporation of sterically demanding units into the P2 position of chemical inhibitors[17,26]. Indeed, this selectivity filter has been used in the development of several DPP8/9- vs. DPP4-selective inhibitors; further structural optimizations may then even lead to inhibitors with a distinct DPP8 vs. DPP9 selectivity[27]. Most inhibitors developed so far are substrate analogs with a reversible or irreversible molecular mode-of-action, with Val-boroPro (VbP, also known as PT-100 or Talabostat) and its derivatives[28–30], 1G244 and analogs, in particular 1G244-12m (Tominostat)[31,32] or Vildagliptin-derived inhibitors as most prominent examples[33]. Also irreversible covalent inhibitors have been described, e.g., a phosphonate-based inhibitor with however unknown proteome-wide target selectivity or the recently by us reported 4-oxo-β-lactam-based DPP8/9 inhibitors[27,30]. A limitation of most DPP8/9 inhibitors is however a lack of knowledge of their proteome-wide selectivity, i.e., information on potential off-targets. For example, the inhibitor 1G244 that is widely used due to its assumed DPP8/9 selectivity, is known to display cytotoxicity due to an unknown off-target mechanism[34].

In this work, we explore alternative starting structures for the design of more selective DPP8/9 inhibitors. In this context, we investigate the natural product Sulphostin (1, Fig. 1b). This structurally unique natural product was originally isolated from a culture broth of *Streptomyces sp.* MK251-43F3 as a potent DPP4 inhibitor (IC$_{50}$ of 21 nM)[35,36]. Sulphostin bears an (S)-3-aminopiperidin-2-one moiety functionalized with a phosphosulfamate group ($R_P$-configuration)[37]. Sulphostin underwent intense structure-activity relationship studies that identified structural determinants for potent inhibition[36]. To test if Sulphostin may also serve as a starting structure for the development of DPP8/9 inhibitors, we chemically resynthesize and test this natural product in diverse biochemical, chemoproteomic and structural

assays, thereby revealing Sulphostin as a covalent inhibitor of DPP4 with the (S)-3-aminopiperidin-2-one moiety serving as a leaving group. From these observations, we then develop second-generation N-phosphonoamide-based DPP8/9-targeted inhibitors with improved inhibitory potency, proteome-wide target selectivity and target engagement in live cells.

## Results
### Sulphostin covalently inhibits DPP4/8/9
Sulphostin has been reported as a potent DPP4 inhibitor[36]. Due to the strong active site homologies between DPP4 and DPP8/9 (Fig. 1a), we investigated if this compound also serves as a starting structure for DPP8/9 inhibitor design. Accordingly, we re-synthesized the natural product following essentially published procedures (Supplementary Fig. 1). We then performed biochemical enzyme inhibition assays using the substrate Gly-Pro-7-amido-4-methylcoumarin (GP-AMC), which releases fluorescent AMC upon enzymatic cleavage, to quantify the inhibitory effect of Sulphostin on DPP8/9 and DPP4 as a positive control. Sulphostin indeed inhibited all three DPP proteins, however with different IC$_{50}$s (Table 1 and Supplementary Fig. 2): DPP4 was most potently inhibited with an IC$_{50}$ value of $79 \pm 29$ nM and thus in a similar range to the reported IC$_{50}$ of 21 nM[36]. The IC$_{50}$ for DPP9 was with $1392 \pm 108$ nM approximately 20-fold higher, while DPP8 was inhibited with an IC$_{50}$ of $6930 \pm 620$ nM. As a point-of-reference and further positive controls, we also tested the established inhibitors VbP, 1G244 and 4-oxo-β-lactam compound 6 (4OβL-6) in the same assay. Again, in accordance with literature, VbP inhibited DPP4/8/9 with IC$_{50}$ values lower than 10 nM[34], while 1G244 displayed DPP8/9 vs. DPP4-selective inhibition, inhibiting DPP8 with an IC$_{50}$ of $4.4 \pm 1.5$ nM and DPP9 with an IC$_{50}$ of $14 \pm 2$ nM[38]. The 4OβL-6 inhibited DPP8 in the nanomolar range and DPP4/9 in the low micromolar range with selectivities comparable to the previous study[27]. Altogether, these results showed that Sulphostin is an inhibitor of DPP4/8/9, although being less potent than established inhibitors.

We then performed native mass spectrometry (nMS) experiments to complement the IC$_{50}$ values with K$_d$ values. Surprisingly, incubation of DPP9 with 50 μM Sulphostin for 90 min resulted in an increase of DPP9's molecular weight by 155 Da and 311 Da instead of the expected increase by 272 Da corresponding to Sulphostin's molecular weight (Fig. 1c and Supplementary Fig. 3). This observed mass difference pattern matches with a covalent modification of dimeric DPP9, at either one or both active sites, with the phosphosulfamate moiety of Sulphostin. Such a modification can be achieved if the phosphotriamide moiety acts as an electrophilic warhead for the active site serine and the (S)-3-aminopiperidin-2-one moiety as a leaving group, resulting in overall covalent inhibition.

With these results in mind, we performed time-dependent DPP4/8/9 biochemical inhibition assays, as reported for other covalent DPP inhibitors (Fig. 1d)[30]. The resulting k$_{obs}$ vs. inhibitor concentration curve had the characteristic shape for an irreversible inhibition mode and was used to calculate the corresponding K$_I$, k$_{inact}$ as well as k$_{app}$ values. In agreement with the previous IC$_{50}$ measurements, these assays confirmed with a k$_{app}$ of 36,963 M$^{-1}$ s$^{-1}$ the preferential binding of Sulphostin to DPP4, followed by DPP9 with a k$_{app}$ of 317 M$^{-1}$ s$^{-1}$ and finally DPP8 with a k$_{app}$ of 11 M$^{-1}$ s$^{-1}$. In summary, the nMS experiments as well as time-dependent enzyme kinetic assays suggest that Sulphostin acts as a covalent DPP4/8/9 inhibitor resulting in a phosphosulfamate modification of the active site of these enzymes.

### Structural analysis of sulphostin's binding mode to DPPs
To gain more detailed insights into the molecular mode-of-action between Sulphostin and the DPP proteins, including a verification of the proposed phosphosulfamate modification of DPPs after Sulphostin application, we solved the structure of Sulphostin-soaked DPP9 crystals (Fig. 2a). With a resolution of 1.89 Å, the structure clearly

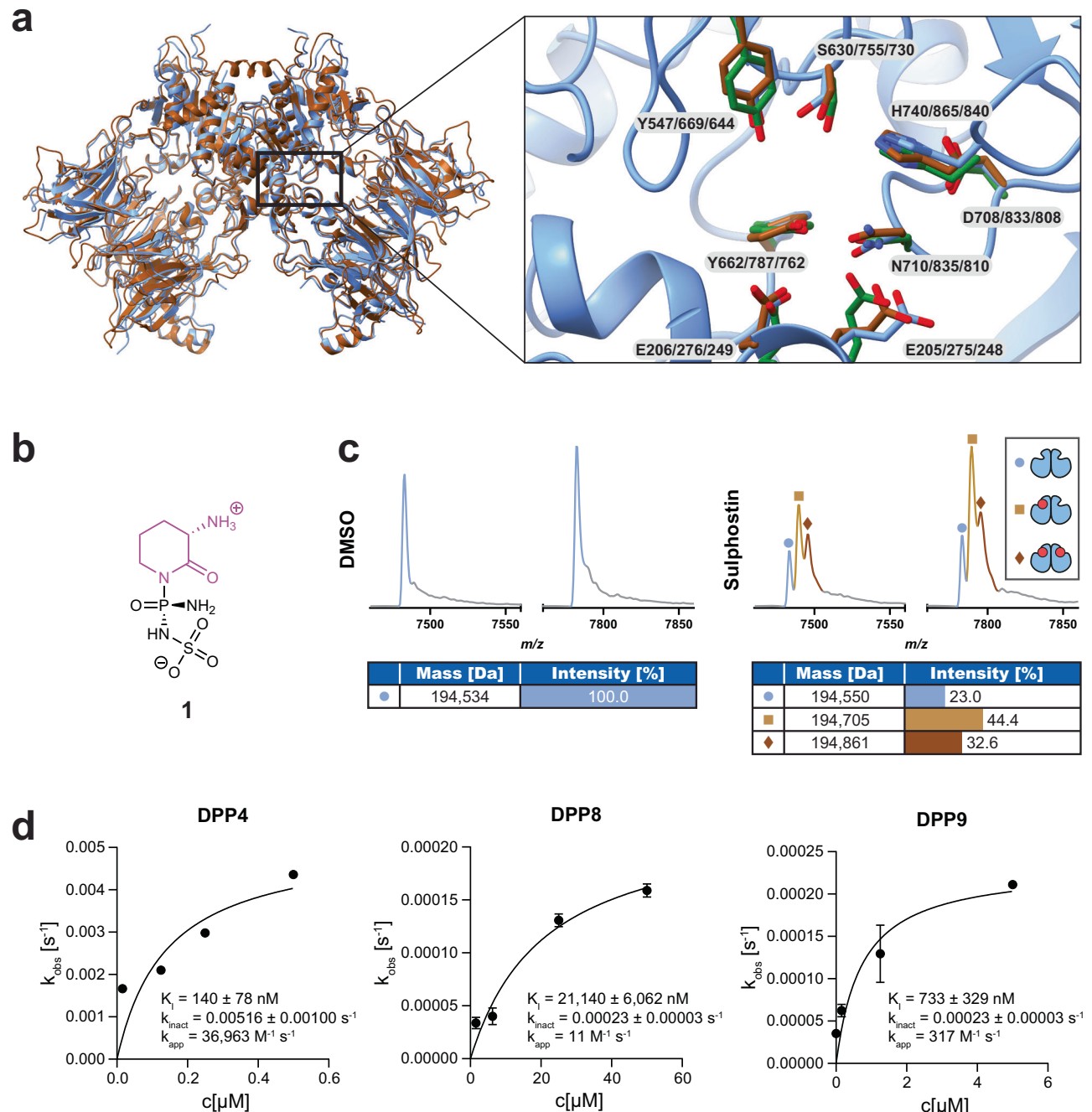

**Fig. 1 | Sulphostin binds to DPP proteins. a** The crystal structures of DPP8 and 9 in the apo state in an overlay, with DPP9 shown in blue and DPP8 in brown as a ribbon model. The active site was zoomed in, showing the active site residues of DPP4 (green), DPP8 (brown) and DPP9 (blue) as stick model. Backbone of DPP9 is shown in blue as ribbon model. Crystal structures from PDB: 1PFQ (DPP4), 6EOO (DPP8) and 6EOQ (DPP9). **b** Chemical structure of the natural product Sulphostin (**1**) containing a phosphosulfamate functional group (indicated in black) and the (S)−3-aminopiperidine-2-one group in purple. **c** Native MS spectrum of DPP9 in the dimeric state with or without Sulphostin bound. Peaks correspond to different charge states. The charge state distribution for the dimer without inhibitor bound is shown in blue, whereas the distribution for the DPP9 dimer with one Sulphostin molecule is shown in light brown and for two Sulphostin molecules in dark brown. **d** Kinetic analysis of DPP inhibition by Sulphostin. The pseudo-first order rate constant ($k_{obs}$) was calculated from an exponential regression of progress curves and plotted against the inhibitor concentration. $K_I$ and $k_{inact}$ were obtained via fitting to a hyperbolic equation. All activity measurements were performed in triplicate ($n = 3$ technical replicates), mean values are shown and error bars indicate the standard error of the mean (SEM). Source data are provided as a Source Data file.

confirmed a phosphosulfamate modification of the active site serine S730 after Sulphostin treatment. No further unassigned electron density was present in the active site, supporting on the one hand the leaving group character of the (S)-3-aminopiperidine-2-one moiety and on the other hand an overall quantitative modification reaction. Importantly, the nucleophilic attack was accompanied by a Walden inversion of the stereochemistry at the chiral phosphorus atom, indicating that the nucleophilic substitution follows an $S_N2(P)$-like mechanism. The modification of the active site serine by the phosphosulfamate moiety also changes the overall orientation of the serine which is now pointing towards the S2-S1 sub-sites. In addition to the covalent interaction, the phosphosulfamate moiety forms hydrogen bonds with the DPP9 residues Y644, Y762, H840, N810, R133 and with the main-chain atoms of Y731. Moreover, ligand interactions with the

**Table 1 | IC$_{50}$ values [nM] for Sulphostin and the reference compounds VbP, 1G244 and 4OβL-6 inhibiting DPP4/8/9**

| Compounds | IC$_{50}$ [nM] | | |
|---|---|---|---|
| | DPP4 | DPP8 | DPP9 |
| Sulphostin | 79 ± 29 | 6930 ± 620 | 1392 ± 108 |
| VbP | 8.3 ± 0.9 | 1.7 ± 0.1 | 1.3 ± 0.1 |
| 1G244 | >100,000 | 4.4 ± 1.5 | 14 ± 2 |
| 4OβL-6 | 8354 ± 728 | 582 ± 36 | 1180 ± 102 |

side chains of E248, E249, Y762, and H840 via water-mediated hydrogen bonds are also observed. Overall, the structure is in full agreement with a covalent phosphosulfamate modification of the active site serine.

We next asked whether this covalent binding mode is unique for DPP9 or also applies to DPP4 and whether the previously hypothesized non-covalent binding mode demonstrates the initial binding event with subsequent formation of a covalent bond[36]. We therefore also determined an X-ray structure of DPP4 after incubation with Sulphostin, which confirmed the observations made with DPP9 (Fig. 2b).

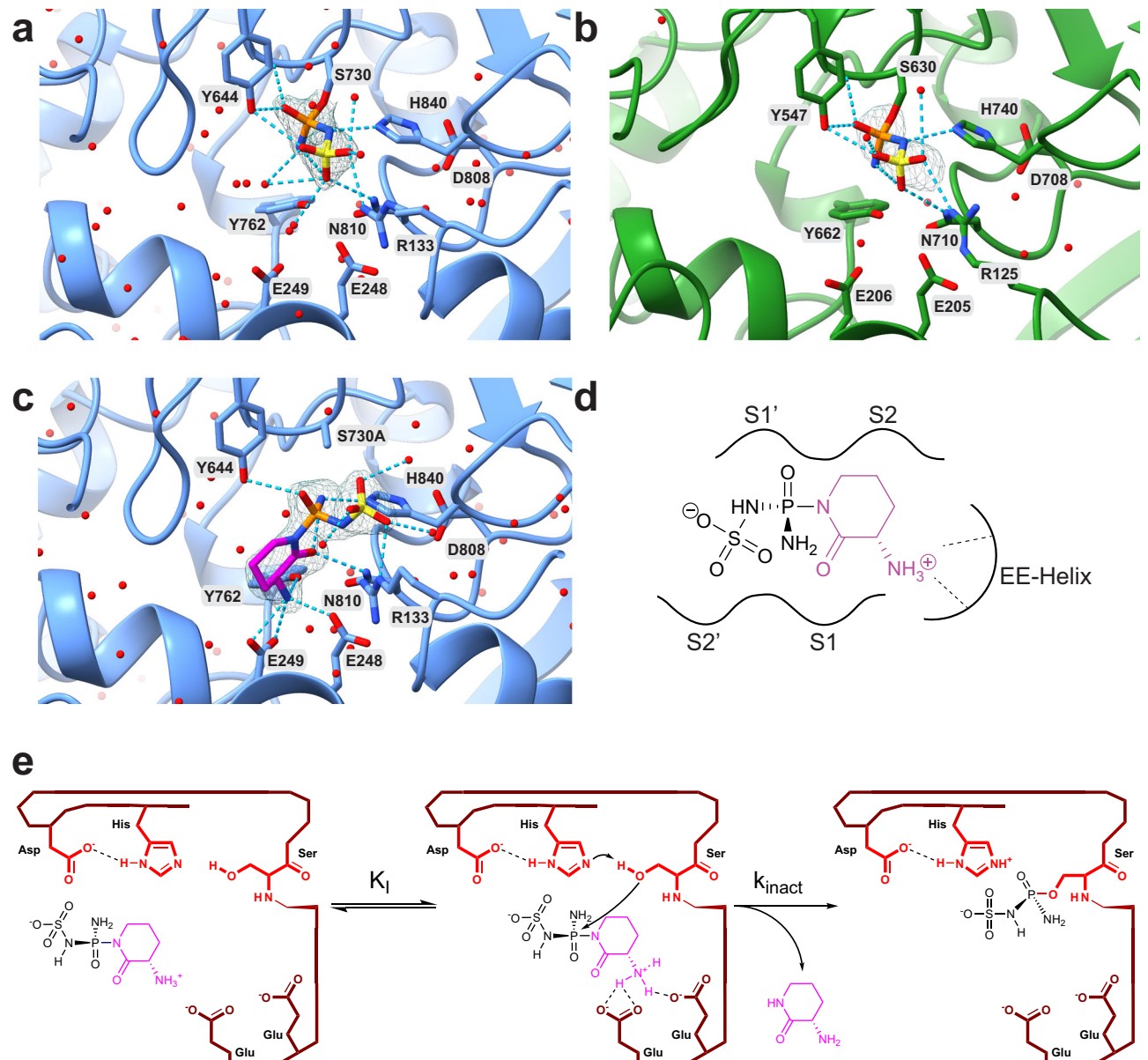

**Fig. 2 | Sulphostin inhibits DPP proteins by a covalent binding mode. a** X-ray crystal structure of Sulphostin in complex with DPP9 refined to a final resolution of 1.89 Å. The phosphosulfamate group is covalently bound to the catalytically active serine S730 residue and is contoured at 1 σ with the refined 2F$_o$-F$_c$ electron density map. Hydrogen bond interactions are depicted as dashed lines. **b** X-ray crystal structure of Sulphostin in complex with DPP4 refined to a final resolution of 2.38 Å. The phosphosulfamate group is covalently bound to S630 and is contoured at 1 σ with the refined 2F$_o$-F$_c$ electron density map. Hydrogen bond interactions are depicted as dashed lines. **c** X-ray crystal structure of Sulphostin in complex with the S730A-DPP9 mutant (1.89 Å resolution) showing the non-covalent interactions prior to the covalent bond formation. The ligand is superimposed with the refined 2F$_o$-F$_c$ electron density map contoured at 1 σ. Hydrogen bond interactions are depicted as dashed lines. **d** Sulphostin is coordinated by E248 and E249 from the EE helix with the free N-terminus of the (S)−3-aminopiperidine-2-one, which occupies two substrate binding sub-sites. **e** The proposed general binding mechanism of Sulphostin with DPP proteins. The stereogenic phosphorus of Sulphostin is attacked by the serine of the catalytic triad, whereby the (S)−3-aminopiperidin-2-one moiety functions as a leaving group.

Again, covalent modification of DPP4's active site serine S630 with a phosphosulfamate group, the product of a nucleophilic attack of the active site serine coupled to a Walden inversion, was exclusively observed, thereby indicating a general covalent reaction mechanism of Sulphostin with DPPs.

Covalent enzyme modification by affinity labels is usually a two-step process. First, a reversible, non-covalent enzyme-inhibitor complex is formed, followed by a second step consisting of the covalent modification reaction (characterized by $K_I$ and $k_{inact}$, respectively). To elucidate the structural determinants underlying the formation of the reversible enzyme-inhibitor complex, we solved the X-ray structure of Sulphostin in complex with a catalytically dead DPP9 S730A mutant (Fig. 2c). In this experiment, a much larger electron density matching the complete structure of Sulphostin was found in the active site, confirming that the reversible enzyme-inhibitor complex is formed with a structurally intact Sulphostin residue and the (S)-3-aminopiperidin-2-one moiety is released only after nucleophilic attack by the active site serine. Sulphostin adopts a substrate-like binding mode, occupying both the S2 and S1 sub-sites with the aminopiperidine-2-one moiety, and with the amino group of the (S)-3-aminopiperidine-2-one forming a hydrogen-bonded ion pair to the glutamates E248 and E249 of the EE-helix. Of note, E248 and E249 line the end of the substrate binding tunnel and typically bind the N-terminus of substrate proteins (Fig. 2d). The (S)-3-aminopiperidine-2-one is thus not a "simple" proline mimic but structurally optimized to span two substrate binding sub-sites and to interact, in analogy to natural substrates, with substrate binding residues in the EE helix. The phospho-acylamide (P = O)-N-(C = O) substructure of Sulphostin, which has comparable chemical reactivities as phosphate anhydrides, however, is critical for activation of the phosphate residue for nucleophilic attack as well as to convert the (S)-3-aminopiperidine-2-one into a leaving group. Of note, the analysis of the structure also revealed an elaborate hydrogen bonding pattern between Sulphostin and DPP9, which matched the observed hydrogen bonds in the covalently conjugated wild-type DPP9 structure.

Overall, these structural analyses not only confirmed a covalent irreversible binding mode of Sulphostin but also enabled to propose a reaction mechanism underlying covalent modification (Fig. 2e). After forming a non-covalent enzyme-inhibitor complex, the active site serine attacks the phosphotriamide moiety, most probably via a $S_N2(P)$ mechanism. Of note, in natural peptidic DPP substrates, the P1'-P2' position is cleaved off upon formation of the acyl intermediate with P1-P2 (Supplementary Fig. 4). In contrast, in case of Sulphostin, the (S)-3-aminopiperidine-2-one moiety corresponding to the P1-P2 position overtakes an important role as a selectivity filter and serves as a good leaving group, whereas the phosphosulfamate corresponding to the P1'-P2' position is covalently attached to the catalytic serine.

## Sulphostin displays promising proteome-wide selectivity

We next wanted to address whether the identified covalent modification reaction is specific for DPPs or if Sulphostin acts as a broadband serine hydrolase inhibitor which would limit its applicability in selective covalent probe design. We therefore performed a competitive activity-based protein profiling (ABPP) with the serine hydrolase-specific activity-based probe (ABP) fluorophosphonate-alkyne (≡FP)[39] (Fig. 3a). HEK293 cell lysates were pre-incubated with 50 μM Sulphostin or DMSO vehicle control for 2 h, followed by a labeling reaction with 2 μM ≡FP. After biotin linkage via click chemistry and streptavidin-based affinity enrichment, bound proteins were on-bead digested with trypsin, followed by LC-MS/MS analysis. The enrichment of each identified protein group was quantified using spectral intensity-based relative quantification using the DMSO-treated samples as a reference. Only identified protein groups with a log2-fold change (FC) of ≥ 2 and a p-value of ≤ 0.01 were considered as primary hits. The employment of ≡FP significantly enriched 21 protein groups, 19 of them serine

hydrolases, including DPP8/9 (Fig. 3b, for a complete list of identified proteins, see Supplementary Data 1), which is slightly less than other comparable studies[40,41], probably due to the application of a click probe, a low probe concentration and an LFQ-based quantification. DPP4 was not detected in our dataset, since it is not expressed in HEK293 cells. Despite the application of a rather high concentration of 50 μM Sulphostin, this pre-treatment significantly competed only the labeling of DPP8/9, suggesting a high target selectivity of Sulphostin within the identified ≡FP target proteins and revealing Sulphostin as a promising starting structure for further inhibitor development.

## Sulphostin-inspired N-phosphono-(S)-3-aminopiperidine-2-ones as DPP inhibitors

Despite its selectivity in its target class, a limitation of Sulphostin as a starting structure for DPP8/9 inhibitor design is its highly polar character, severely limiting membrane permeability and thus the possibility to inhibit intracellularly localized DPP8/9 under physiological conditions. With the knowledge of the importance of the (S)-3-aminopiperidine-2-one leaving group for inhibition potency and selectivity, we therefore rationally designed Sulphostin-inspired N-phosphono-(S)-3-aminopiperidine-2-one derivatives to 1) improve cell membrane permeability, 2) increase potency and selectivity towards DPP8 or DPP9, and 3) obtain Sulphostin derivatives featuring an alkyne handle, thus allowing direct target identification studies to evaluate proteome-wide target selectivities (Fig. 4a).

The designed phosphono-(S)-3-aminopiperidine-2-ones (which can also be denoted as 3-amino-1-[alkyl (ethoxy)phosphoryl]piperidin-2-ones) were chemically synthesized through the activation of phosphonate diethyl ester derivatives (which were either commercially obtained or prepared as reported in Supplementary Fig. 5a) with oxalyl chloride (Fig. 4b and Supplementary Fig. 5b), followed by a nucleophilic substitution on the resulting chloro(ethoxy)phosphoryl intermediate with in situ lithiated Alloc-protected (S)-3-aminopiperidine-2-one. Alloc cleavage with catalytic Pd(PPh3)4, then yielded the desired Sulphostin-inspired N-phosphono-(S)-3-aminopiperidine-2-one derivatives (Table 2).

In order to obtain cell-permeable inhibitors, the N-phosphono-(S)-3-aminopiperidine-2-one was synthesized with hydrophobic residues at the phosphonate residue. The first hydrophobic residues were structurally simple alkyl chains of different lengths (for chemical structures with the corresponding inhibition values, see Table 2 and Supplementary Fig. 6). Biochemical inhibition assays with these compounds revealed that longer alkyl chains resulted in lower IC50 values for DPP8/9, while DPP4 inhibition was basically invariant to alkyl chain length, suggesting a simple selectivity filter for achieving DPP8/9 selectivity vs. DPP4. The derivative with a C12 alkyl chain (compound 8) showed with an IC50 of 96 ± 8 nM for DPP8 and 486 ± 49 nM for DPP9 the strongest inhibition. The derivatives with a C8 and C10 alkyl chain were slightly less potent, nevertheless still nanomolar DPP8 inhibitors with however a ten-fold better DPP8 vs. DPP9 selectivity (compound 6 and 7).

In order to obtain chemical probes for direct target identification, two derivatives with an alkyne handle were synthesized (compounds 9 and 10). In contrast to the observed inhibition trend for alkyl derivatives, the derivative with the shorter C5-alkyne residue (compound 9) inhibited DPP8 with an IC50 of 668 ± 43 nM and DPP9 with an IC50 of 1573 ± 92 nM. Compound 9 was thus about 20-fold more potent than the corresponding derivative with a longer C8-alkyne chain (compound 10). Indeed, the C5-alkyne derivative 9 is approximately 20-fold more potent inhibitor than alkyl compound 5, which features an alkyl chain of comparable length.

This observation led to the hypothesis that a π system located three carbons downstream of the warhead might be favorable for inhibition. Accordingly, a set of derivatives with these structural features were synthesized and tested in biochemical assays (compounds

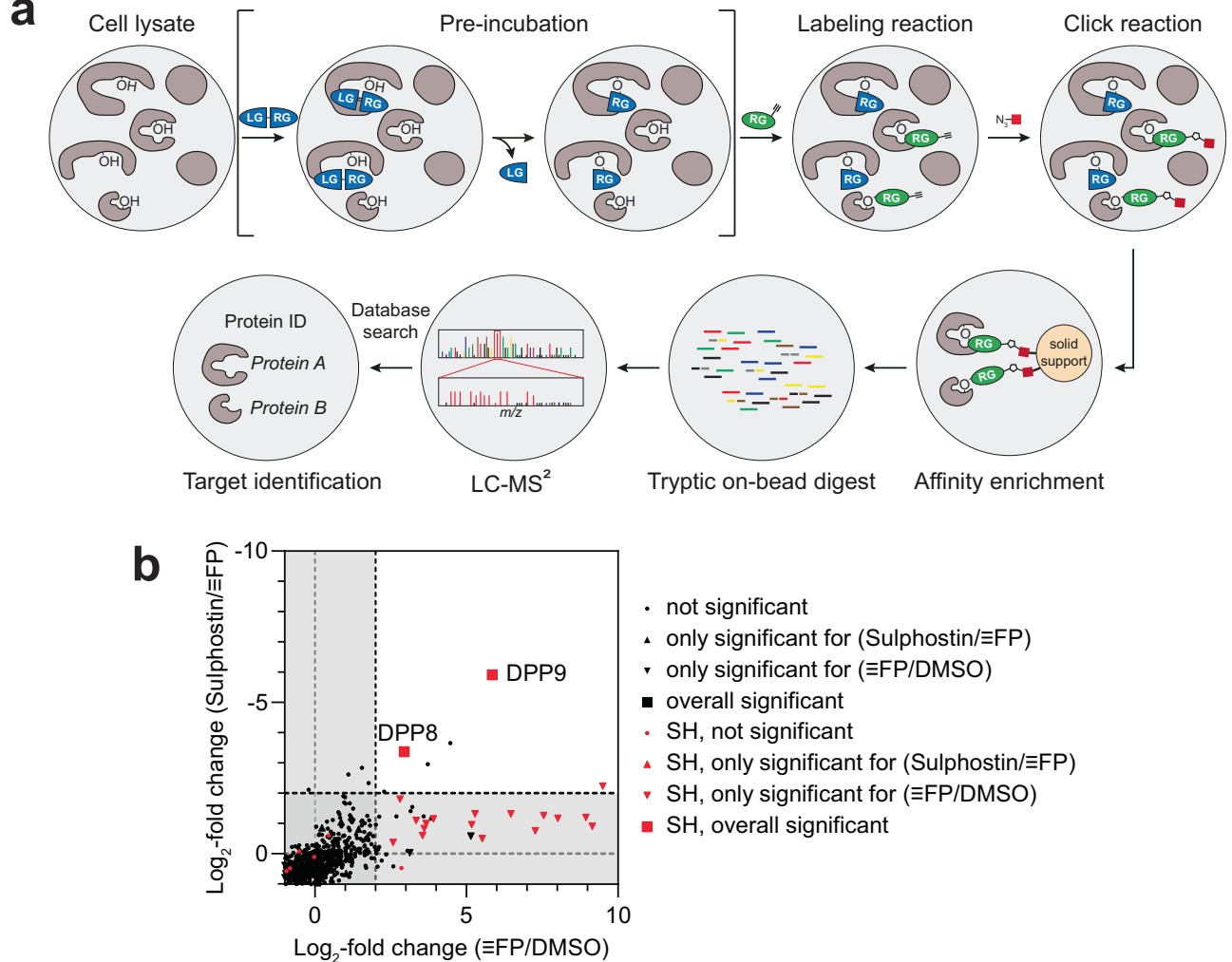

**Fig. 3 | Sulphostin shows promising selectivity within FP target proteins.**
**a** Overview of a competitive ABPP workflow. A cell lysate is pre-incubated with an inhibitor, composed of a leaving group (LG) and a reactive group (RG), followed by a labeling reaction with a probe that covalently binds target proteins via its RG, which can be either an ABP such as FP or an alkyne-tagged ligand. After click attachment of a biotin moiety, labeled proteins are enriched via solid support, tryptic digested on-bead, and subsequently identified via LC-MS/MS analysis. **b** Cell lysates were treated with 50 μM Sulphostin or DMSO vehicle control prior to treatment with the ABP FP. ABPP of SHs without (FC plotted on x-axis) or after pre-treatment with 50 μM Sulphostin (competition experiment, FC plotted on y-axis) with 2 μM ≡FP after click chemistry ($n = 4$ biologically independent samples). Dashed lines indicate the FC ≥ 2 (≡FP/DMSO) or FC ≤ -2 (Sulphostin/≡FP) threshold. Black symbols indicate identified protein groups, red symbols highlight serine hydrolases (SHs). To identify statistically significant hits from the analysis (marked as a triangle or square), $p$-value ≤ 0.01 (two-sided Student's $t$-test, permutation-based FDR with 250 randomizations and FDR = 0.05) was applied. Source data are provided as a Source Data file.

**11, 12, 13, 14, 15**). The introduction of a phenyl ring (compound **11**) indeed resulted in more potent DPP8/9 inhibition (IC$_{50}$ of 370 ± 23 nM and IC$_{50}$ of 1909 ± 158 nM, respectively), although the selectivity against DPP4 was reduced. Introduction of a bromo substituent at the *para-* (compound **12**) or *meta-*position (compound **13**) resulted in further improvements in potency and selectivity, with the *para-*substitution providing better inhibition values. A sulfone derivative **15**, and its precursor **14**, were also prepared in an attempt to mimic the hydrogen bonding capabilities of the sulfamate moiety in Sulphostin (for their synthesis, see Supplementary Figs. 5b and 7). The sulfone group (**15**) however turned out to be detrimental for inhibition selectivity and potency, while the introduction of a thioether (**14**) was better tolerated. In an attempt to combine the inhibition features of the aromatic system with the observed improved inhibition of longer alkyl chains, compound **16** was synthesized and showed the strongest inhibition of DPP8 with an IC$_{50}$ of 14 ± 1 nM and best selectivity over DPP9 ( ~21-fold) and DPP4 ( ~1154-fold) of all tested derivatives. Analogous to Sulphostin, we then investigated the time-dependent

inhibition of DPP4/8/9 for all synthesized compounds, resulting, as before, in a characteristic hyperbolic dependence of k$_{obs}$ on the inhibitor concentration in presence of a sufficient inhibition of enzyme activity, indicating an irreversible modification (Supplementary Fig. 8 and Supplementary Table 1). The rate constants for irreversible inactivation were consistent with IC$_{50}$ values, proving the high potency and selectivity of **16**, as a k$_{app}$ of 20 M$^{-1}$ s$^{-1}$ was determined for DPP4, 51,249 M$^{-1}$ s$^{-1}$ for DPP8 and 2310 M$^{-1}$ s$^{-1}$ for DPP9 (Fig. 4c). Overall, these studies show that proper combination of a classical phosphonate warhead decorated with an (*S*)-3-aminopiperidine-2-one leaving group leads to potent DPP inhibitors with promising selectivity.

**Structural analysis of the binding mode of *N*-phosphono-(*S*)-3-aminopiperidine-2-ones**
To verify that the synthesized derivatives share the same binding mode as the parent compound Sulphostin and to elucidate the structural basis of their inhibition potency, a crystal structure of the most potent *N*-phosphono-(*S*)-3-aminopiperidine-2-one, compound **16**, and DPP9

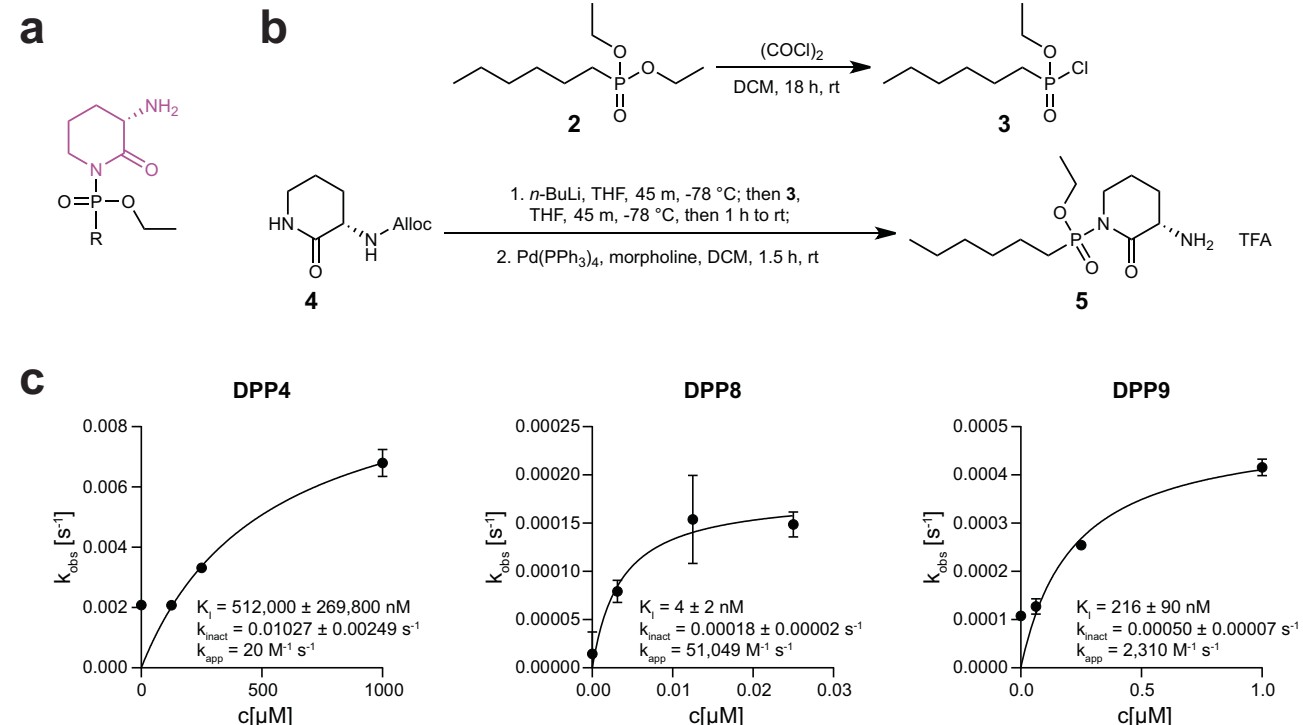

**Fig. 4 | Sulphostin-inspired *N*-phosphono-(*S*)−3-aminopiperidine-2-ones inhibit DPPs. a** Chemical structure of Sulphostin-inspired *N*-phosphono-(*S*)−3-aminopiperidine-2-ones. The previously identified leaving group is shown in purple. **b** Representative synthesis route of *N*-phosphono-(*S*)−3-aminopiperidine-2-ones by activation of a phosphonate diethyl ester **2** with oxalyl chloride yielding a chloro-phosphonate intermediate **3**, which then reacts with in situ lithiated Alloc-protected (*S*)−3-aminopiperidine-2-one **4** resulting in corresponding *N*-phosphono-(*S*)−3-aminopiperidine-2-one **5**. **c** Kinetic analysis of DPP inhibition by *N*-phosphono-(*S*)−3-aminopiperidine-2-ones. The pseudo-first order rate constant ($k_{obs}$) was calculated from an exponential regression of progress curves and plotted against the inhibitor concentration. $K_I$ and $k_{inact}$ were obtained via fitting to a hyperbolic equation. All activity measurements were performed in triplicate ($n = 3$ technical replicates), mean values are shown and error bars indicate the SEM. Source data are provided as a Source Data file.

was analyzed (Fig. 5a). Analogous to Sulphostin, the electron density revealed a covalent modification of the catalytically active serine residue (S730) of DPP9, this time with a corresponding 2-(4-pentylphenylthio)ethylphosphonate moiety, confirming that the (*S*)-3-aminopiperidine-2-one moiety also here acts as a leaving group. Although an epimeric mixture of (3*S*, *R*$_P$) and (3*S*, *S*$_P$) diastereomers of **16** was added, the electron density showed that the phosphorus atom is exclusively in an *S*$_P$ configuration after nucleophilic attack by the active site serine, indicating that the active diastereomer has the (3*S*, *R*$_P$)-configuration, again matching with the spatial arrangement found in Sulphostin. The ethoxy group of the phosphonate binds to the hydrophobic S1 sub-site formed by V756, V811 and Y762. The 2-(4-pentylphenylthio)ethyl residue extends towards the S' region and then bends sharply to protrude into the S2 pocket. Specifically, the pentyl tail occupies the S2 pocket, however, it appears highly flexible due to the lack of strong specific interactions with the protein. The whole binding mode is further stabilized by a specific hydrogen bond between the phosphoryl oxygen and the Y644 side chain. Overall, the structural analysis shows that the *N*-phosphono-(*S*)-3-aminopiperidine-2-ones are indeed structural mimics of Sulphostin but with higher biochemical inhibition properties and different DPP target selectivities.

### *N*-phosphono-(*S*)-3-aminopiperidine-2-ones display improved proteome-wide target selectivities and target engagement in cells

To gain insight into the proteome-wide target selectivity of *N*-phosphono-(*S*)-3-aminopiperidine-2-ones, the alkyne-tagged derivative compound **9** was used in a direct covalent target protein identification approach. We initially performed a competitive ABPP experiment in

HEK293 cell lysates via pre-incubation with a 5-fold excess of compound **8**, the non-tagged structural analog **11** and compound **16**. The LC-MS/MS analysis after application of **9** led to the significant enrichment of only two proteins, namely DPP9 and the off-target prolyl endopeptidase (PREP), another analog of the DASH family which is very often not evaluated in DPP8/9 inhibitor development studies (Fig. 5b, for a complete list of identified proteins, see Supplementary Data 2). Both proteins were significantly competed by pre-incubation with **11**, whereas solely DPP9 was competed by **8** and **16**. To quantify inhibition of the off-target PREP, we performed additional PREP biochemical inhibition assays using the same experimental set up as for DPP4/8/9 (Supplementary Table 2 and Supplementary Fig. 9). Compounds **9** and **11** inhibited PREP, however only with an IC$_{50}$ value of 13,418 ± 4040 nM and 2207 ± 1577 nM, respectively. More importantly, our most potent compound **16** displayed no activity against PREP within the nanomolar and low micromolar range ( < 50 μM), demonstrating the high potential of the Sulphostin-inspired *N*-phosphono-(*S*)-3-aminopiperidine-2-ones for the development of selective DPP inhibitors.

After these promising results, we next investigated target engagement of compounds **8** and **16** in cells. To this end, we measured whether their application disturbs the interaction between DPP9 and its substrate BRCA2. We recently showed that DPP9 plays a role in the repair of DSBs via regulation of the formation of RAD51 foci after DPP9-mediated BRCA2 cleavage[24]. The interaction between DPP9 and BRCA2 is induced by the presence of DSBs, for example by the genotoxic agent Mitomycin C (MMC). These interactions can be visualized and quantified in cells using proximity ligation assays (PLAs), where each dot represents a single interaction event between DPP9 and BRCA2. Importantly, fewer MMC-induced PLA events between DPP9 and

**Table 2 | IC$_{50}$ values [nM] for different N-phosphono-(S)−3-aminopiperidine-2-one compounds inhibiting DPP4/8/9**

| R | IC$_{50}$ [nM] | | |
|---|---|---|---|
| | DPP4 | DPP8 | DPP9 |
| **5** C6-alkyl | 46,671 ± 6534 | 13,662 ± 735 | 42,669 ± 2551 |
| **6** C8-alkyl | 25,903 ± 5307 | 664 ± 115 | 6994 ± 1279 |
| **7** C10-alkyl | 19,961 ± 4300 | 231 ± 15 | 2125 ± 1236 |
| **8** C12-alkyl | 28,219 ± 3709 | 96 ± 8 | 486 ± 49 |
| **9** C5-alkyne | 2989 ± 970 | 668 ± 43 | 1573 ± 92 |
| **10** C8-alkyne | >100,000 | 11,418 ± 1033 | 32,686 ± 4630 |
| **11** | 952 ± 40 | 370 ± 23 | 1909 ± 158 |
| **12** | 243 ± 8 | 65 ± 3 | 179 ± 14 |
| **13** | 942 ± 55 | 199 ± 18 | 630 ± 14 |
| **14** | 4451 ± 174 | 2016 ± 118 | 13,250 ± 886 |
| **15** | 4713 ± 201 | 5864 ± 584 | 15,888 ± 652 |
| **16** | 16,150 ± 1350 | 14 ± 1 | 298 ± 21 |

BRCA2 are observed following addition of 1G244 (Fig. 5c and Supplementary Fig. 10)[24], a competitive DPP8/9 inhibitor that induces a closed conformation of the active site[17]. Notably, a significant reduction in the number of MMC-induced PLA events for DPP9 with BRCA2 is also observed by the addition of 10 μM of compounds **8** and **16** prior to cell fixation. These results strongly suggest that, like 1G244, compounds **8** and **16** compete with BRCA2 for access to the active site of DPP9.

To test whether this direct target engagement leads to a corresponding phenotype, we examined whether these compounds increase the sensitivity of cells to MMC, similar to the hyper-sensitivity observed in DPP9 depleted (DPP9$^{KO}$) cells[24]. Previous studies have shown that at high concentrations or prolonged application, DPP8/9 inhibitors can result in an overall reduced cell viability due to off target effects[42–44]. Consistently, treatment of HeLa wild-type cells with 1G244 (10 μM) for 72 h, coincided with a reduced cell viability (Supplementary Fig. 11a). Lower cell viability was also observed in response to compounds N-phosphono-(S)-3-aminopiperidine-2-ones **7** and **8**, while compound **16** did not show cytotoxic effects under these conditions (Supplementary Fig. 11a).

We then tested the effect of DPP8/9-targeting compounds on cell viability after co-treatment with 1 μM of MMC and either the selected N-

phosphono-(S)-3-aminopiperidine-2-ones, 1G244 or Sitagliptin, a highly selective DPP4 inhibitor (Fig. 5d and Supplementary Fig. 11b)[45]. As a positive control, we analyzed the viability of HeLa DPP9$^{KO}$ cells in response to MMC, which showed the expected hyper-sensitivity compared to that of wild-type cells, consistent with published data[24]. In line with the higher sensitivity of DPP9$^{KO}$ cells to MMC, 1G244 significantly reduced cell viability of HeLa wild-type cells, while the DPP4 inhibitor Sitagliptin had no additional effect after MMC treatment. Importantly, similar to 1G244, also N-phosphono-(S)-3-aminopiperidine-2-ones **8** and **16** clearly increased the sensitivity of cells to MMC, with a reduced cell viability. Here, compound **16** stands out since, in contrast to 1G244, it shows less off-target effects (Fig. 5d and Supplementary Fig. 11a). In summary, our experiments confirm N-phosphono-(S)-3-aminopiperidine-2-ones as promising target selective DPP8/9 inhibitors, which display direct target engagement in cells.

## Discussion

Covalent probes are highly promising chemical tools and potential starting points for drug discovery. A persisting challenge however is the development of target (class) selective inhibitors, due to a lack of systematic design strategies. Such strategies have in part been

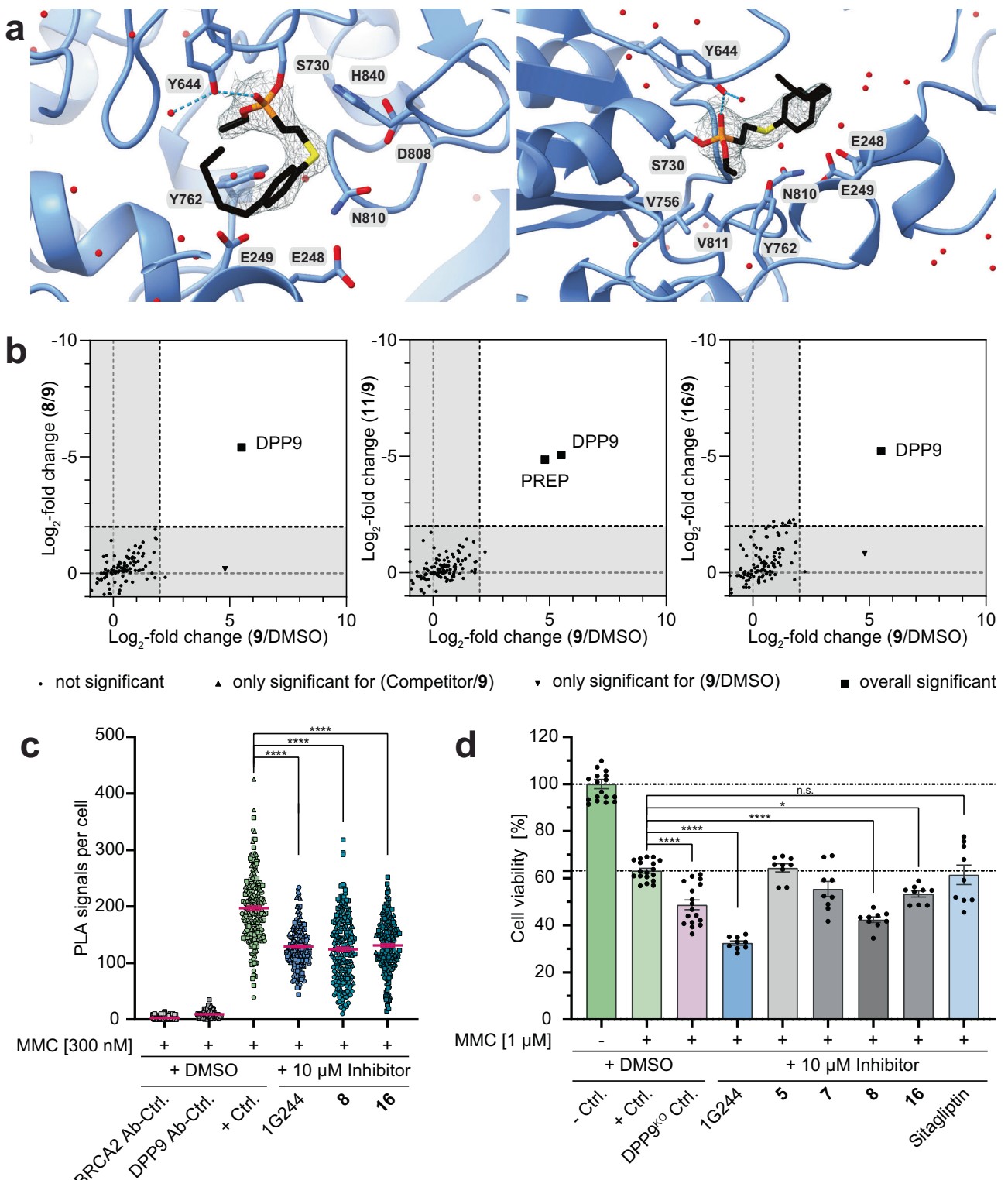

b

not significant     ▲ only significant for (Competitor/9)     ▼ only significant for (9/DMSO)     ■ overall significant

developed for kinases for which covalent targeting of cysteines located in proximity of the active site has become an effective strategy for achieving target selectivities[1,46]. For serine proteases, such a concept is however not yet available.

Here, we have shown that DPPs, a clinically relevant class of serine exoproteases, can be selectively targeted by phosphonate-based probes via the incorporation of a "simple" (S)-3-aminopiperidine-2-one leaving group as a selectivity filter. This design is different from classical phosphonate-based serine hydrolase inhibitor strategies in

which selectivity is usually achieved by optimizing the structural region of the inhibitor that remains covalently bound to the enzyme after reaction, while the leaving group is in most cases structurally comparatively simple and does not afford target selectivity. The previously reported diaryl isoindolin-1-ylphosphonate-based DPP8/9 inhibitors are a prime example for this design type[30]. A systematic survey on the use of leaving groups for achieving target selectivity has, to the best of our knowledge, not yet been undertaken, although we anticipate that for example, the previously reported 1,2,3-triazole

**Fig. 5 | *N*-phosphono-(*S*)−3-aminopiperidine-2-ones bind covalently to DPP9.**
**a** X-ray structure of **16** in complex with DPP9 refined to a final resolution of 2.49 Å, shown from two different angles. The ligand is covalently bound to the catalytically active serine S730 residue and is contoured at 1 σ with the refined 2F$_o$-F$_c$ electron density map. Hydrogen bond interactions are depicted as dashed lines. **b** ABPP without (FC plotted on x-axis) or after pre-treatment with 50 μM **8**, **11** or **16** (competition experiment, FC plotted on y-axis) with 10 μM **9** after click chemistry (*n* = 4 biologically independent samples). Dashed lines indicate the FC ≥ 2 (**9**/DMSO) or FC ≤ -2 (Competitor/**9**) threshold. Black symbols indicate identified protein groups. To identify statistically significant hits from the analysis (marked as a triangle or square), *p* ≤ 0.01 (two-sided Student's *t*-test, permutation-based FDR with 250 randomizations and FDR = 0.05) was applied. **c** Addition of *N*-phosphono-(*S*)−3-aminopiperidine-2-ones block the interaction between endogenous DPP9 and BRCA2. Quantification of PLAs showing MMC-induced DPP9-BRCA2 PLA events in HeLa wild-type cells, in the presence of 10 μM of the respective inhibitory compounds. Cells were treated with 300 nM MMC for 24 h and 10 μM of the indicated inhibitors for 1 h prior to fixation. Each dot represents the number of PLA events in a

single cell. Data were analyzed by unpaired two-sided *t*-test comparisons (****$p$ < 0.0001). To visualize the three biological replicates, each biological replicate is labeled differently: circle, triangle or square. *n* values indicate total number of cells in each analysis group from 3 independent biological replicates in total: *n* = 240 for BRCA2 Ab-Ctrl., *n* = 237 for DPP9 Ab-Ctrl., *n* = 222 for +Ctrl., *n* = 202 for 1G244, *n* = 231 for **8**, *n* = 205 for **16**. **d** Inhibition of DPP8/9 increases cellular sensitivity to genotoxic stress. HeLa wild-type (WT) cells were treated with 1 μM MMC and 10 μM of the respective inhibitors. Cell viability was measured after 72 h, and normalized to WT mock treated cells. DPP9$^{KO}$ cells and WT cells treated with 1G244 (10 μM) were analyzed as positive controls, whereas WT cells treated with Sitagliptin were used as a negative control. Dashed lines indicate the viability of the WT + MMC cells. The graph shows the mean and error bars indicate the SEM of all individual measurements (*n* = 18 (HeLa wild-type controls), 17 (HeLa DPP9$^{KO}$ control) or 9 (inhibitor-treated samples) independent biological replicates). Data were analyzed by a Tukey two-way ANOVA, using a mixed effect analysis (n.s. = not significant, *$p$ = 0.0157, ****$p$ < 0.0001). Source data are provided as a Source Data file.

ureas may represent an inhibitor structure class in which the chemical structure of the leaving group may be a decisive factor for the observed target selectivities[47].

The development of our design strategy was initiated by the serendipitous finding that the natural product Sulphostin acts as an irreversible covalent DPP4/8/9 inhibitor. Our work is thus not only an example on how natural products can serve as an inspirational source for drug design[48], it is also an example on how minor changes in the evolution of natural products can result in highly diverse mode-of-actions. Notably, Phaseolotoxin and its cleavage product Octicidin, the most important representative of the so far only other known sulphophosphotriamide natural product class, acts in contrast to Sulphostin as a reversible, non-covalent transition state inhibitors of ornithine carbamoyltransferase (Supplementary Fig. 12)[49]. We propose that the observed difference in reactivity between Phaseolotoxin and Sulphostin stems from the presence of an amine in Phaseolotoxin vs. an amide moiety in the (*S*)-3-aminopiperidine-2-one substructure of Sulphostin, resulting in the case of Sulphostin in the generation of a reactive warhead prone to nucleophilic attack by an active site serine residue.

The Sulphostin-inspired *N*-phosphono-(*S*)-3-aminopiperidine-2-ones analogs are not only an alternative warhead for serine hydrolase inhibition but also display enhanced inhibition potency and selectivity as well as target engagement in cells. They thus represent valuable chemical tools that can complement existing DPP8/9 chemical probes for usage in diverse biomedical research settings and they may serve as starting points for drug discovery. In addition, the highly selective mode-of-action could enable the design of more advanced chemical probes, e.g., quenched activity-based probes (qABPs) for monitoring DPP8/9 activity in cells[50]. In addition, we anticipate that activated *N*-phosphono(*S*)-3-aminopiperidine-2-ones may, due to their favorable reactivity profile as weak electrophiles, find wider application in the design of covalent probes in the future.

## Methods
### Chemical synthesis
Detailed information on the chemical synthesis and the synthetic procedures of all compounds can be found in the Supplementary Information.

### Biochemical assays
Enzyme activity of DPP4, DPP8, DPP9 (Proteros Biostructures GmbH, Germany) and PREP (Sigma Aldrich, USA) was determined by measuring the initial velocity of AMC release from the fluorogenic substrate GP-AMC (Sigma Aldrich, USA) using a Tecan Spark® 10 M multimode microplate reader (Tecan Group Ltd., Switzerland) in activity buffer (20 mM HEPES/KOH pH 7.3, 110 mM potassium acetate, 2 mM Mg acetate, 0.5 mM EGTA, 0.02% Tween 20, supplemented with

1 mM DTT) and a final volume of 20 μL for maximal 60 min at 30 °C. Prior to activity measurement, 10 nM enzyme was incubated for 45 min at 30 °C and gentle shaking with the inhibitor compounds in concentrations ranging from 0 to 100 μM. The compounds were dissolved in DMSO and diluted in activity buffer (final DMSO concentration: 1%). To the inhibitor-enzyme mixture, GP-AMC was added to a final concentration of 250 μM for DPP4, 8, and 9 or 2 mM for PREP. Afterward, enzyme activity was determined by calculating the slope (fluorescence release over time) and plotted against the concentrations of the inhibitor. IC$_{50}$ values were calculated using Graph-Pad Prism v10.2.3 using the equation: [Inhibitor] vs. response−Variable slope (four parameters). All measurements were performed in triplicates and error bars represent standard errors of the fit.

To calculate the reaction rate of covalent bond formation between DPP proteins and inhibitor, k$_{inact}$, K$_I$ and k$_{app}$ was determined as previously described[30,51]. Therefore, 250 μM GP-AMC was added simultaneously with a serial dilution of the inhibitor to 10 nM enzyme. Enzyme activity was measured for a maximum of 120 min at 30 °C, and fluorescence intensities were then plotted against time. Data was fitted to the following equation: $Y = (vi/kobs)*(1 − \exp(−kobs*X))$ using Graph-Pad Prism v10.2.3. The obtained pseudo-first-order rate constant was used to determine the maximum enzyme inhibition rate (k$_{inact}$) and binding affinity (K$_I$) by fitting the plot of k$_{obs}$ against inhibitor concentration to the following equation: $Y = kinact/(1 + (KI/[I]))$. The efficiency of covalent bond formation (k$_{app}$) was calculated by the ratio of k$_{inact}$/K$_I$. All measurements were performed in triplicates and error bars represent standard errors of the fit.

### In vitro competitive ABPP
For ABPP experiments, HEK293 cells (Cytion, Germany) were cultured in Dulbecco's modified Eagle's medium (DMEM, Gibco®, USA) supplemented with 10% fetal calf serum (Gibco®, USA) and 1% Penicillin/Streptomycin (Gibco®, USA) at 37 °C under a humidified 5% carbon dioxide atmosphere. Cells were cultured in 10 cm tissue culture dishes. A sub-cultivation ratio of 1:3 to 1:5 was used for passaging cells every 2–3 days when cells reached approximately 80-90% confluence.

All probes and competitors were dissolved in DMSO. To examine selectivity of Sulphostin and *N*-phosphono-(*S*)-3-aminopiperidine-2-ones within a proteome, cell lysates were generated from HEK293 cells grown on a cell 10 cm tissue culture dish. Therefore, cells were washed with phosphate buffered saline (PBS, Gibco®, USA) and detached with trypsin. Cells were then pelleted via centrifugation (5 min, 250 × *g*, RT), followed by three consecutive wash steps with PBS. Cell pellets were re-suspended in 400 μL phosphate buffer (50 mM HNa$_2$PO$_4$, pH 8) and lysed by sonication (Bioruptor UCD-200, Diagenode, Belgium; 1 min pulse and 0.5 min pause in 10 cycles with high power). The cell lysates were cleared via centrifugation (30 min, 20,817 × *g*, 4 °C) and protein

concentration was determined with Roti®-Nanoquant (modified Bradford assay, Carl Roth, Germany). A total protein amount of 1000 μg was pre-incubated with 50 μM of the respective competitor (1 or 2 h, 37 °C, vigorous shaking) and then labeled with 2-10 μM of the indicated probe (1 h, 37 °C, vigorous shaking). Labeled enzymes were subjected to a click reaction with 10 μM TAMRA-biotin-N₃ (Jena Bioscience, Germany) 100 μM TBTA (Sigma Aldrich, USA), 2 mM TCEP (Sigma Aldrich, USA), and 1 mM CuSO₄ (Sigma Aldrich, USA; 1 h, RT, in the dark).

## Affinity enrichment and MS preparation

After click reaction, proteins were precipitated by a modified MeOH/CHCl₃ precipitation protocol[52]. In brief, protein solutions were incubated with four equivalents of MeOH (overnight, -20 °C), followed by the addition of one equivalent chloroform and three equivalents of MS-grade water (VWR Chemicals, USA). The precipitated proteins were washed thrice with MeOH, dried on air, and dissolved in a final volume of 7.9 mL 0.2% (w/v) SDS in 1× PBS (60 min, gentle shaking, 37 °C). For enrichment of labeled proteins, 100 μL avidin-agarose bead slurry was added to the solution and incubated (1 h, gentle rotation, RT). Enriched proteins were then washed five times with 1% SDS in MS-grade water and thrice with MS-grade water (10 min, gentle rotation, RT). After every wash step, beads were recovered by centrifugation (5 min, 400 × g, RT). Beads were then taken up in 100 μL of 0.8 M Urea in 50 mM ammonium bicarbonate (ABC) and proteins were sequentially reduced with 5 mM dithiothreitol (DTT) in 50 mM ABC (30 min, vigorous shaking, 37 °C) and alkylated with 10 mM iodoacetamide (IAM) in 50 mM ABC (30 min, vigorous shaking, 37 °C, in the dark). After quenching of the alkylation reaction with DTT (final concentration of 10 mM), proteins were digested by adding 1 μg trypsin (16 h, vigorous shaking, 37 °C). Beads were then collected by centrifugation, the supernatant recovered and acidified with formic acid (FA, final concentration of 0.5% (v/v)). The beads were washed by adding 40 μL 1% (v/v) FA (5 min, vigorous shaking, RT) and the supernatant was combined with the previously recovered peptide mixture. To remove remaining beads, the mixture was passed over a homemade two-disc glass fiber membrane StageTip (5 min, 100 × g, RT). Subsequently, peptide mixture was desalted on homemade C18 StageTips as described previously[53]. All centrifugation steps were performed in the range of 600–800 × g and for 2 min at RT. In brief, acidified peptide mixtures were passed over StageTips and immobilized peptides were washed twice with 0.5% (v/v) FA. The peptides were then eluted by a two-step elution with 80% (v/v) acetonitrile (ACN) containing 0.5% (v/v) FA and dried using a vacuum concentrator (Eppendorf, Germany). For LC-MS/MS analysis, peptides were re-suspended in 15 μL 0.1% (v/v) FA.

## LC-MS/MS

The competitive ABPP experiment with Sulphostin was analyzed with an Oribtrap Elite instrument (Thermo Fisher Scientific, USA) coupled to an EASY-nLC 1000 liquid chromatography (LC) system (Thermo Fisher Scientific, USA), whereas the competitive ABPP with *N*-phosphonopiperidones was analyzed with an Orbitrap Fusion Lumos mass spectrometer (Thermo Fisher Scientific, USA) coupled to a Vanquish Neo LC system (Thermo Fisher Scientific, USA). Both LCs were operated in the one-column mode. The analytical column was a fused silica capillary (75 μm × 28–32 cm) with an integrated sintered frit (CoAnn Technologies ICT36007515F-50-5, USA) packed in-house with Kinetex C-18-XB Core Shell 1.7 μm resin (Phenomenex, Germany) and encased by a column oven (Sonation, Germany), whose temperature was adjusted to 50 °C during data acquisition. The analytical column was attached to a nanospray flex ion source (Thermo Fisher Scientific, USA). The LC was equipped with two mobile phases: solvent A (0.2% (v/v) FA, 2% (v/v) ACN, in water) and solvent B (0.2% (v/v) FA, 20% (v/v) H₂O in ACN). All solvents were of ultra-high-performance liquid chromatography (UHPLC) grade (Honeywell, Germany). Peptides were directly loaded onto the column with a maximum flow rate that would

not exceed the pressure limit of 980 bar and subsequently separated on the analytical column using different gradients (105 or 140 min length; for details, see Supplementary File Sample_Legend_and_LC-MS_Settings, Section "LC_Settings")

The Orbitrap Elite mass spectrometer was operated using Xcalibur software v3.0.63 and the Orbitrap Fusion Lumos mass spectrometer using Xcalibur software v4.7.69.37 in the positive ion mode. Precursor ion scanning (MS¹) was performed in the Orbitrap analyzer (FTMS; Fourier Transform Mass Spectrometry with the internal lock mass option turned on (lock mass was 445.120025 *m/z*, polysiloxane))[54]. Dynamic exclusion was enabled (exclude after *n* times = 1; Exclusion duration (s) = 30; mass tolerance = ±10 or 100 ppm). MS² product ion spectra were recorded only from ions with a charge bigger than +1 and in the data-dependent fashion in the ITMS (Ion Trap Mass Spectrometry). All relevant individual settings (resolution, scan rate, scan range, AGC, ion acquisition time, charge states, isolation window, fragmentation type and details, cycle time, number of scans performed, and various other settings) for the individual experiments can be found in Supplementary File Sample_Legend_and_LC-MS_Settings, Section "MS_Settings".

## Protein identification and statistical analysis

Raw spectra were submitted to an Andromeda search in MaxQuant (version 2.0.3.0 or 2.5.2.0) using the default settings[55]. Label-free quantification as well as match between runs was activated. The MS/MS spectra were searched against the Uniprot *H. sapiens* reference database (Homo_sapiens.fasta, 79684 entries) and a contaminants database (implemented in MaxQuant, 246 entries) that contains known MS contaminants to estimate the level of contamination. Andromeda search allowed the static modification of a cysteine (57 Da, alkylation with IAM) as well as the oxidation of methionine residues (16 Da) and acetylation of the protein N-terminus (42 Da) as dynamic modifications. Enzyme specificity was set to "Trypsin/P". The instrument type in Andromeda searches was set to Orbitrap and precursor mass tolerance was set to ±20 ppm (first search) and ±4.5 ppm (main search). The MS/MS match tolerance was set to ±0.5 Da. The peptide spectrum matched FDR and the protein FDR was set to 0.01 (based on target-decoy approach). The minimum peptide length was seven amino acids. For protein quantification, unique and razor peptides were allowed. Modified peptides with dynamic modifications were allowed for quantification and the minimum score for modified peptides was 40. Match between runs was enabled with a match time window of 0.7 min and match ion mobility window of 0.05 min[56]. Further data analysis and filtering of the MaxQuant output was done in Perseus (version 2.0.3.0)[57]. Therefore, LFQ-intensities were loaded into the matrix from the proteinGroups.txt file and potential contaminants, hits from the reverse database as well as hits only identified by peptides with a modification site were removed. Related biological replicates were combined into categorical groups to allow comparison of different treatments. LFQ-intensities were log₂-transformed and only those protein groups that were found in at least three of four replicates in at least one categorical group were kept for further analysis. Missing values were then imputed from a normal distribution (width 0.3, downshift 1.8) and protein enrichment by respective ABPs was calculated based on a two-sided Student's *t*-test (permutation-based FDR: 0.05, s0 = 0, 250 randomizations) compared to the DMSO control. To examine the effect of the competitor pretreatment on protein enrichment, a two-sided Student's *t*-test (permutation-based FDR: 0.05, s = 0, 250 randomizations) was performed to calculate the difference in protein abundance between noncompetitive and pretreated probe-labeled samples and the statistical significance of the fold change. For both approaches, the log₂-fold change (FC) of probe-labeled samples was plotted against FC of the pretreated probe-labeled samples using Graph-Pad Prism v10.2.3. Protein groups with a FC ≥ 2 (Probe/DMSO), FC ≤ −2 (Competitor/Probe), and *p*-value ≤ 0.01 were considered as primary hits.

## Cell viability assay

HeLa wild-type (HeLa Flp-In T-REx WT, kind gift from Prof. Matthias Hentze) and HeLa DPP9$^{KO}$ cells (prepared in-house) were cultured in DMEM (ThermoFisher, USA) supplemented with 10% FBS (PAN Biotech, Germany), 2 mM L-Glutamine (ThermoFisher, USA) and 1% Penicillin/Streptomycin (100 U mL$^{-1}$ Penicillin and 100 µg mL$^{-1}$ Streptomycin, ThermoFisher, USA). Cells were seeded in 96-well flat-bottom cell culture plates (Greiner, Germany) at a concentration of $5 \times 10^3$ cells per well (37 °C and 5% CO$_2$). The indicated concentration of Mitomycin C (MMC, Merck) was added to the cells 24 h after seeding, together with 10 µM of the indicated inhibitor. The inhibitors were refreshed every 24 h. Control cells were treated with DMSO.

Cell viability was measured 72 hours after the MMC treatment. For this, the medium was removed and replaced with phenol-red free DMEM (ThermoFisher, USA) containing 0.5 mg mL$^{-1}$ MTT (Sigma Aldrich, USA). Cells were incubated for 30 min at 37 °C, 5% CO$_2$. Subsequently, the medium was removed and replaced by DMSO. The plates were shaken on an orbital shaker for 20 min before measuring the MTT signal on a CARIOstar Plus microplate reader (BMT labtech, Germany). Each experiment was performed three times, each in triplicates.

## Proximity ligation assay

HeLa wild-type cells were cultured as described above. Cells were seeded on coverslips in 24-well plates (Greiner, Germany) at a concentration of $2 \times 10^4$ cells per well (37 °C and 5% CO$_2$). To induce a DPP9-BRCA2 interaction, 300 nM MMC was added to the cells 8 h after seeding and for a total incubation time of 24 h. Indicated Inhibitors (10 µM) were added 1 h prior to cell fixation. Cells were fixed with 4% formaldehyde in PBS for 10 min and permeabilized with 0.2% Triton-X-100 in PBS for 5 min. The proximity ligation assay (PLA) was performed using the Duolink in situ PLA Kit (Sigma Aldrich, USA). Briefly, cells were washed with PBS and blocked with Duolink Blocking solution for 60 min at 37 °C. Cells were incubated with primary antibodies against BRCA2 (R&D Systems, USA, Cat# MAB2476, RRID:AB_2259370, 1:100) and DPP9 (Ruth Geiss-Friedlander−University of Freiburg Cat# RGF_1, RRID:AB_2889071, 1:100) for 90 min at 37 °C and actin filaments were simultaneously counterstained with CytoPainter Phalloidin-iFluor 488 Reagent (Abcam, United Kingdom). Antibody control coverslips were treated with only one primary antibody to estimate background staining in each experiment. Coverslips were washed and treated with PLA reagents according to the manufacturers protocol. Cells were mounted in DAKO with DAPI fluorescent mounting medium and analyzed using an LSM 710 confocal microscope, 63x NA 1.4 oil immersion objective (Zeiss, Germany). Images were processed using Zen black v2.3 (Zeiss, Germany) and subsequently analyzed using the Duolink ImageTool Setup 1.0.1.2 (Sigma Aldrich, USA).

## Native MS

Purified recombinant DPP9 (theoretical molecular weight: 97,266 Da) was exchanged to 200 mM ammonium acetate buffer (NH$_4$OAc, pH 6.8; Sigma-Aldrich, USA; diluted in MS water; Honeywell, Germany). For this purpose, a rebuffering procedure using 10 kDa molecular weight cut-off spin-filter columns (Merck Millipore, USA) was applied for five cycles[58]. The resulting protein concentration was determined using microvolume spectroscopy (DS-11+, DeNovix, USA) and adjusted to 5 µM with 200 mM NH$_4$OAc. To the protein solution 50 µM of Sulphostin or the respective solvent control (2.5% DMSO) was added and the mixture was incubated for 90 min.

The samples were ionized using a TriVersa NanoMate nanoESI system (Advion, USA) equipped with 5 µm diameter nozzle spray chips (Advion, USA). A 5 µL sample volume was picked out of 96-well plates and the ESI spray was generated using 0.8 psi nitrogen backpressure combined with a positive nozzle chip voltage of 1.7 kV with spray sensing turned on (15 s threshold). A total number of 1000 MS spectra were recorded in positive EMR mode on an Exactive Plus EMR Orbitrap mass spectrometer (Thermo Fisher Scientific, USA) calibrated with CsI (2 mg mL$^{-1}$; Thermo Fisher Scientific, USA). The MS parameters used are as follows: $m/z$ = 5000–10,000; capillary temperature [°C] = 150; microscan count = 1; FT resolution 17,500; AGC target = 1*10$^6$; HCD = 200; isCID = 200; UHV sensor [mbar] = 5.8*10$^{-10}$–6.0*10$^{-10}$.

UniDec software (version 6.0.4) was used for mass deconvolution[59]. For this purpose, the default settings were used within an $m/z$ range of 6500 to 9500, with the exception of a higher deconvolution resolution (sample mass every 1 Da), a narrower peak detection window (peak detection range of 80 Da) and a total peak normalization. GraphPad Prism v8.0.1 was used for data visualization.

## X-ray crystallography

C-terminally His-tagged human DPP9 20-863 wildtype and S730A were expressed and purified as described previously[17]. DPP9 wildtype and mutant apo crystals were obtained by hanging drop vapor diffusion method mixing protein solution (11 mg ml$^{-1}$ in 20 mM TRIS-HCl pH 8, 150 mM NaCl, 2 mM DTT) with reservoir solution (20% (w/v) PEG2000MME, 0.1 M TRIS-HCl pH 6.75) in a 1:1 ratio at 293 K. For complex formation with **1** apo crystals were soaked with 5 mM ligand for 2 h. Before flash cooling in liquid nitrogen crystals were cryo protected by immersing them in reservoir solution supplemented with 20% (v/v) Ethylene glycol.

Co-crystals of wildtype DPP9 in complex with ligand **16** were obtained by hanging drop vapor diffusion by mixing protein solution (11 mg ml$^{-1}$ in 20 mM TRIS-HCl pH 8, 150 mM NaCl, 2 mM DTT + 5 mM **16**) with reservoir solution (20% (w/v) PEG2000MME, 0.1 M TRIS-HCl pH 7.25) in a 1:1 ratio at 293 K. Before flash cooling in liquid nitrogen crystals were cryo protected by immersing them in reservoir solution supplemented with 20% (v/v) Ethylene glycol.

C-terminally His-tagged human DPP4 39-766 was expressed and purified as described previously[60]. Co-crystals of DPP4 in complex with ligand **1** were obtained by hanging drop vapor diffusion method mixing protein solution (18 mg ml$^{-1}$ in 20 mM HEPES-NaOH pH 7.3, 100 mM NaCl + 2 mM **1**) with reservoir solution (24% (w/v) PEG2000MME, 0.1 M BICINE pH 9.00, 0.16 M Li$_2$SO$_4$) in a 1:1 ratio at 293 K. Crystals were flash cooled in liquid nitrogen.

Diffraction data were collected at PXII/X10SA (SLS, Villigen, Switzerland) for DPP4:**1** (wavelength: 1.0000 Å, temperature: 100 K), DPP9:**1** (wavelength: 0.9998 Å, temperature: 100 K) as well as DPP9 Ser730Ala:**1** (wavelength: 1.0000 Å, temperature: 100 K) and at ID23-1 (ESRF, Grenoble, France) for the DPP9:**16** (wavelength: 0.8856 Å, temperature: 100 K). Data were processed using the programs autoPROC, XDS (BUILT = 20200417 or 20230630) and autoPROC, AIMLESS (v0.7.7)[61]. Structures were solved by molecular replacement by phaser (v2.8.3)[62] using previously solved structures of DPP9 (PDB ID: 6EOQ) and DPP4 (PDB ID: 1PFQ) as search models. Subsequent model building and refinement were performed according to standard protocols with COOT (v0.8.9.2 or v0.9.8)[63] and REFMAC (v5.8.0267, v5.8.0430 or v5.8.0419)[64], respectively. The ligand parameterization and generation of the corresponding library files were carried out with CORINA (v4.4.00026)[65]. A summary of data collection and refinement statistics for each structure is provided in Supplementary Table 3. Images of **1** or **16** in complex with DPP4/8/9 were made with UCSF ChimeraX v1.7.1[66]. The structural datasets have been deposited in the ProteinDataBank[67] with the following PDB accession codes: 9GOH (DPP4:**1**, Ramachandran statistics: favored: 95.25%, allowed: 4.75%, and outliers: 0.00%), 9GON (DPP9:**1**, Ramachandran statistics: favored: 97.26%, allowed: 2.62%, and outliers: 0.12%), 9GOC (DPP9 Ser730Ala:**1**, Ramachandran statistics: favored: 97.41%, allowed: 2.47%, and outliers: 0.12%), and 9GOD (DPP9:**16**, Ramachandran statistics: favored: 96.49%, allowed: 3.33%, and outliers: 0.19%).

## Reporting summary

Further information on research design is available in the Nature Portfolio Reporting Summary linked to this article.

## Data availability

The structural datasets generated in this study have been deposited in the Protein Data Bank repository[67] under the accession codes 9GOH (DPP4:**1**) [https://doi.org/10.2210/pdb9goh/pdb], 9GON (DPP9:**1**) [https://doi.org/10.2210/pdb9gon/pdb], 9GOC (DPP9 Ser730Ala:**1**) [https://doi.org/10.2210/pdb9goc/pdb], and 9GOD (DPP9:**16**) [https://doi.org/10.2210/pdb9god/pdb]. The crystal structures used in this study are available in the Protein Data Bank repository[67] under the accession codes 1PFQ (DPP4) [https://doi.org/10.2210/pdb1pfq/pdb], 6EOO (DPP8) [https://doi.org/10.2210/pdb6eoo/pdb] and 6EOQ (DPP9) [https://doi.org/10.2210/pdb6eoq/pdb]. The mass spectrometry proteomics data have been deposited to the ProteomeXchange Consortium via the PRIDE[68] partner repository with the dataset identifier PXD058896. The processed chemoproteomic data are available in Supplementary Data 1 and 2. Source data are provided with this paper.

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

## Acknowledgements

This work was supported by the Deutsche Forschungsgemeinschaft CRC1430, project ID 424228829 to M. K., F. K. and D. H., and GRK 2606, project ID 423813989 to R. G.-F. and S. Z.; R. G.-F. acknowledges the networking support of the ProteoCure COST action (CA20113). Microscopy images were made in the Lighthouse Core Facility which is funded in part by the Medical Faculty, University of Freiburg (Project Number 2021/B3-Fol).

## Author contributions

R.G.-F., D.H., R.H., and M.K. conceived, designed and supervised the study. W.W.A.T., L.F., C.J.A.V. contributed to compound and synthetic route design, conducted the chemical synthesis and characterization. L.S. and M.N. performed and analyzed enzyme assays as well as ABPP experiments. A.L. performed crystallography. S.Z. conducted and analyzed cell viability and proximity ligation assays. L.S., D.P. and F.K. performed MS and related data analysis. L.S., W.W.A.T., L.F. and M.K. wrote the manuscript.

## Funding

## Competing interests

A.L. is an employee of Proteros Biostructures GmbH. The remaining authors declare no competing interests.
