## [Transparent Peer Review file · Nature Communications]

Sulphostin-inspired N-phosphonopiperidones as selective covalent DPP8 and DPP9 inhibitors

Corresponding Author: Professor Markus Kaiser

Version 0:

Reviewer comments:

Reviewer #1

(Remarks to the Author)

This study identifies a natural product (sulphostin) as an irreversible inhibitor of several dipeptidyl-peptidases, provides x-ray crystallography-based structural insight into the irreversible binding mode, and delivers new analogues of sulphostin with an increased selectivity for dipeptidyl-peptidases 8/9 (DPP8/9). The study also claims to demonstrate target engagement in cellular experiments. Finally, it suggests that sulphostin analogues can be regarded as selective dipeptidyl-peptidase inhibitors with respect to other proteases that are present in cellular proteomes.

Sulphostin was reported in 2001 as a dipeptidyl-peptidase 4 (DPP4) inhibitor. It was not investigated as an inhibitor of other DPPs at the time. In at least one follow-up paper of the 2001 report (Abe et al. *Bioorg. Med. Chem.* ref. 2005, reference 36 in this manuscript), new analogues of sulphostin were also reported. Contrary to what the authors seem to suggest (bottom of page 6), Abe et al. did not report sulphostin to be a reversible inhibitor. To the best of my knowledge, Abe et al. did not report experimental findings about the inhibition type, they only added a fairly low-level docking pose of a sulphostin derivative to their manuscript. If this would be viewed as an initial binding mode, the docking pose would not even be contradictory with a subsequent irreversible step. I think the authors should therefore correct their statement and mention that Abe et al did not report experimental data about the inhibition type. Likewise, the authors should also add to the introduction that other relevant irreversible inhibitors for DPP8/9 have been reported recently (Carvalho et al. *Angew. Chem. Int. Ed.* 2022, e202210498), many of the authors of that report also figure on this one.

Overall, the submitted manuscript contains research of high quality. The findings described in it could have relevance, but I also have a number of questions about this work that would need to be addressed (see below). In the absence of the answers to these questions, this is definitely a high-level chemical/structural biology report, but it is very hard to judge the full potential impact of this work, especially whether the molecules would have value for downstream DPP8/9 research.

1) It is hard to judge the potency of the reported new compounds because (except for sulphostin) only IC₅₀ data are reported in Table 2. IC₅₀ values are acceptable for comparing potencies of the new compounds relatively to each other, but they are not adequate for irreversible inhibitors per se. I also feel that the incubation time of the inhibitors with the enzymes prior to activity determination is rather long (45 min.): this typically 'inflates' IC₅₀-potencies for irreversible compounds. Therefore, it is my opinion that the authors should report the rate constants for irreversible inactivation, at least for their most potent DPP8/9 inhibitors. Alternatively, they could include IC₅₀ data for an irreversible reference inhibitor, for example one of the Carvalho compounds, under the conditions used in this study. This could help to maximally assess the relevance of the new compounds in this paper based on potency.

2) The use of cell viability as a read-out for target engagement seems very tricky. The data in Figure 5c do not unequivocally demonstrate target engagement. In my personal opinion, the lower viability observed with compounds 7, 8 and 16 might well be the consequence of intrinsic cytotoxicity rather than of DPP9 inhibition. After all, the inhibitor concentration used in this experiment (10 μM) is rather high and off-target effects can reasonably be anticipated at this concentration. To amend this situation, at least the effect on viability should be shown for the compounds in the same cells, but not in the presence of mitomycin. Also, dose-dependence of the effect should be investigated. Very honestly, I would actually prefer to see a direct measure of target engagement if this would be within the means of the team. Finally, it is somewhat disturbing that the concentration of reference 1G244 in these studies, is not mentioned.

Reviewer #2

(Remarks to the Author)

The work reported by the authors is of great interest to the chemical biology and drug discovery communities. Their demonstration of the covalent nature of sulphostin towards DPP opens new horizons, which the authors have already started to investigate with a convincing series of N-phosphonopiperidones.

To demonstrate selectivity beyond DPP4/8/9, the authors have utilised chemoproteomics. I have been asked by the editor to specifically review this part of the manuscript.

Two experiments are being reported:

- 1) one studying the target space of Sulphostin using a general fluorophosphonate-alkyne serine hydrolase probe (results in figure 3) and
- 2) one studying the target space of cpd 11 using tailored probe 9

The experimental part indicates how the raw data were processed before being plotted. The supplementary tables give the list of identified proteins together with the related significance and $\log_2(\text{ratios})$. A table with the initial LFQ values is not, unless I am mistaken, provided. However, because data have been imputed, it would be reasonable to be able to judge how many values for the serine hydrolases have been imputed and how successful is the enrichment (target intensities vs non-target intensities for instance).

In this line of thought, it would be important for the reader to be informed about the success of the enrichment with the fluorophosphonate-alkyne: how many serine hydrolases are usually captured by other groups using this probe and workflow?

Continuing on the first experiment (figure 3), the manuscript indicates that 21 protein groups, mostly serine hydrolases were significantly enriched by the general probe. I believe that all serine hydrolases measured in this experiment should be annotated with another colour on the volcano plot. The protein within the threshold decided by the authors contains 6 proteins where only 2 are annotated: DPP8 and DPP9. It is not clear at first sight that the red square are the only ones that pass a significance criterium, the unannotated red square being therefore PYCR2. This figure would benefit from a legend on the figure, and the authors could consider plotting the size of the dots according to significance rather than a hard arbitrary threshold. Any reader will wonder what is the significance of the 4 remaining proteins in the "target quadrant". Additionally, instead of "competitor" and "probe" their identity should feature in the axis labels.

More importantly, I believe that the claim of selectivity would benefit from an amendment of the experiment seen in 5b. As such, the following elements do not allow for me to be perfectly convinced:

- a) In the first experiment with a general probe (figure 3b) DPP8 can be enriched and competed by sulphostin despite a mediocre IC50. However in the final experiment DPP8 cannot be seen in the expected quadrant despite the better potency of 9 and 11 against DPP8 vs DPP9, the latter being expectedly found. Since this target is missed, it is quite difficult to be convinced that other targets are not also missing from the profile.
- B) Compounds 5, 7, 8 & 16 are being tested in cells, not 11. The selectivity of 11 is thereby of little interest, while the authors put forward 8 and 16 as their favourite molecules.

The chemoproteomics experiment should thereby concern molecules 8 and/or 16 as competitors with 9 as the enrichment probe.

Additionally, as mentioned for figure 3b, the legend of figure 5b could be made more explicit for the sake of clarity.

Compared to figure 3a, also, only the warhead remains (more similarly to ligand-directed chemistry) on the target proteins, rendering the scheme in figure 3a somewhat inaccurate to explain the latter experiment.

Other Minor comments:

1) A few Germanisms have slipped through the authors attention and should be attended to:

- a) a few "phosphor" instead of "phosphorus" in p6
- b) "sulfon" without a "e" p9
- c) "applied" l294

2) The number of significant digits for many values reported in the manuscript are not reasonably representing the estimated experimental errors.

Reviewer #3

(Remarks to the Author)

The main discoveries of the manuscript "Sulphostin-inspired N-phosphonopiperidones as selective covalent DPP8 and DPP9 inhibitors" are (1) Sulphostin was shown to be a covalent inhibitor of DPP4/9, (2) high-resolution crystallography confirmed the binding mode of Sulphostin, (3) proteomic analysis showed that only DPP8/9 binds to Sulphostin

intracellularly, (4) various Sulphostin derivatives were synthesized and compound 16 was found, (5) the complex structure of compound 16 with DPP9 was determined and the compound 16 was shown to be a covalent inhibitor, (6) intracellular targeting tests showed that compounds 9 and 11 bind to DPP9 and PREP, and (7) investigations of the effect of DPP8/9 inhibitors on cell sensitivity to genotoxic agents showed that the tested compounds were promising target selective DPP8/9 inhibitors, which are active in cells. Thus, this research was conducted in a very logical manner, utilizing methods from diverse fields such as biochemistry, physical chemistry, organic chemistry, and cell biology.

As mentioned above, this study has yielded a number of useful data, but the most interesting point is that target selectivity is ensured by the leaving group of the inhibitor. Not only for DPP9, but many covalent inhibitors for serine proteases mimic acyl enzyme intermediates. In such cases, the specificity is ensured by the N-terminal fragment structure that binds to the catalytic Ser, and the structure of the leaving group (the C-terminal fragment of the cleaved peptide, analogous to a peptide structure) has not received much attention.

I appreciate the extensive work carried out by authors which may be useful in developing novel DPP9 inhibitors. I enjoyed reading the manuscript. However, I have some comments that I believe will enhance the clarity and impact of the author's findings.

MAJOR comments

1. Comparing the structure of Sulphostin to the substrate peptide structure, the leaving group of Sulphostin (3-aminopiperidin-2-one moiety) corresponds to the P2-P1 residue of a substrate peptide, and a phosphosulfamate moiety to P1'-P2'. On the other hand, in peptide cleavage by DPP9, P1'-P2' is removed to form an acyl intermediate, P2-P1-OG(Ser730). This difference in reaction mechanism is very interesting from an enzymatic chemistry point of view and should definitely be discussed in more detail. Specifically, please add a figure of the reaction mechanism of acyl intermediate formation in peptide cleavage by DPP9 to Fig. 2 (as Fig. 2f) or provide an additional supplementary figure showing a comparison with Fig. 2e.

2. Have you performed any experiments using the 3R epimer of Sulphostin in the evaluation of DPP9 inhibitory activity or in the structural analysis of its complex with DPP9?

3. Complexes of DPP9 and DPP4 with Sulphostin were obtained by the soaking method, which is reasonable evidence in that Sulphostin is a covalent inhibitor. However, there is a concern that the authors may have failed to detect the structural changes caused by Sulphostin binding due to the limitations by crystal packing. On the other hand, the complex of compound 16 with DPP9 was obtained by co-crystallization, so there is little risk of missing the structural changes caused by inhibitor binding. Therefore, by comparing the structures of the DPP9/compound 16 complex and the DPP9/Sulphostin complex, the authors may be able to check the above concerns in the DPP9/Sulphostin complex structure.

MINOR comments

1. lines 568-569, 574-575 and 580-581, please unify the notations: TRIS-HCl or tris, pH x or pH=x.xx

2. lines 569 and 575, 0.10 Tris -> 0.10 M Tris

3. line 582, a 1:1 ration at 293 K -> a 1:1 ratio at 293 K.

4. In Fig. 1a, the hydrogen bond between the double Glu motif (E248 and E249) and the water molecule occupying the position corresponding to the 3-position nitrogen of the Sulphostin (corresponding to the N-terminal of the substrate peptide) should also be shown.

5. In Tables 1 and 2, the IC50 value of 1G244 for DPP4 seems to be evaluated only at relatively low concentrations (indicated as >100 nM) as compared with other compounds. What happens when tested at higher concentrations? Is it not possible to test at higher concentrations for 1G244 due to solubility issues, etc.?

6. Table S2 (Data Collection), What criteria did you use to determine the upper limit of resolution for the diffraction data? According to Table S2, the CC_half value of the outer shell is well above 0.5 for each of the data, so I would think that a little higher resolution data would be available.

7. Table S2 (Refinement), The size of the test set used for refinement is quite small, less than 1%, what is your aim? Please comment.

Version 1:

Reviewer comments:

Reviewer #1

(Remarks to the Author)

I appreciate the authors' thoughtful replies to my remarks and I also agree with the additional experimental work that they have done and the clarifications they have made in the manuscript. Overall, I feel that my remarks have been adequately addressed and I therefore have no objections that this manuscript is published in Nature Communications.

Reviewer #2

(Remarks to the Author)

I would like to thank the authors for the additional experiments and amendments they made to their manuscript. I am glad to see that my request led to find that the best molecules do not hit PREP. I believe the manuscript is suitable for publication and I wish great impact to the authors.

I would like to add however the following, mainly as food for thoughts: I think that the choice of imputing values this way for this type of experiments does not do justice to the inhibitors. This type of imputation that assumes a normal distribution of the proteins close to the level of detection is valid for full proteomes; is it really for enrichment experiments? I am convinced that DPP8 is bound by the 3 molecules and that 0 values compared to 3 values in the "competition" should be good enough to call it a target. However, I do appreciate the dictatorship of the p-value in the field that leads to use imputation even if doubtful.

Reviewer #3

(Remarks to the Author)

I have read the revised version of Sewald et al. and confirm that my comments on the first version have been adequately addressed in the revised version.

I therefore consider the revised version to be substantially improved and worthy of publication in Nature Communications.

REVIEWER COMMENTS

Reviewer #1 (Remarks to the Author):

This study identifies a natural product (sulphostin) as an irreversible inhibitor of several dipeptidyl-peptidases, provides x-ray crystallography-based structural insight into the irreversible binding mode, and delivers new analogues of sulphostin with an increased selectivity for dipeptidyl-peptidases 8/9 (DPP8/9). The study also claims to demonstrate target engagement in cellular experiments. Finally, it suggests that sulphostin analogues can be regarded as selective dipeptidyl-peptidase inhibitors with respect to other proteases that are present in cellular proteomes.

We thank the reviewer for his/her overall positive feedback on our study.

Sulphostin was reported in 2001 as a dipeptidyl-peptidase 4 (DPP4) inhibitor. It was not investigated as an inhibitor of other DPPs at the time. In at least one follow-up paper of the 2001 report (Abe et al. *Bioorg. Med. Chem.* ref. 2005, reference 36 in this manuscript), new analogues of sulphostin were also reported. Contrary to what the authors seem to suggest (bottom of page 6), Abe et al. did not report sulphostin to be a reversible inhibitor. To the best of my knowledge, Abe et al. did not report experimental findings about the inhibition type, they only added a fairly low-level docking pose of a sulphostin derivative to their manuscript. If this would be viewed as an initial binding mode, the docking pose would not even be contradictory with a subsequent irreversible step. I think the authors should therefore correct their statement and mention that Abe et al. did not report experimental data about the inhibition type. Likewise, the authors should also add to the introduction that other relevant irreversible inhibitors for DPP8/9 have been reported recently (Carvalho et al. *Angew. Chem. Int. Ed.* 2022, e202210498), many of the authors of that report also figure on this one.

We agree that Abe *et al.* never specifically reported Sulphostin as a reversible inhibitor and that the published docking binding mode could also represent the initial step of a subsequent irreversible reaction; also the reported workflows and calculations for the biochemical inhibition assay could be interpreted in such a manner. We therefore carefully went through our manuscript and revised the corresponding statement.

We also extended our introductory statements on irreversible DPP8/9 inhibitors, including the mentioned Carvalho *et al.* study; please note that we had already cited this study in this context (reference 27 in our manuscript).

Overall, the submitted manuscript contains research of high quality. The findings described in it could have relevance, but I also have a number of questions about this work that would need to be addressed (see below). In the absence of the answers to these questions, this is

definitely a high-level chemical/structural biology report, but it is very hard to judge the full potential impact of this work, especially whether the molecules would have value for downstream DPP8/9 research.

1) It is hard to judge the potency of the reported new compounds because (except for sulphostin) only IC₅₀ data are reported in Table 2. IC₅₀ values are acceptable for comparing potencies of the new compounds relatively to each other, but they are not adequate for irreversible inhibitors per se. I also feel that the incubation time of the inhibitors with the enzymes prior to activity determination is rather long (45 min.): this typically 'inflates' IC₅₀-potencies for irreversible compounds. Therefore, it is my opinion that the authors should report the rate constants for irreversible inactivation, at least for their most potent DPP8/9 inhibitors. Alternatively, they could include IC₅₀ data for an irreversible reference inhibitor, for example one of the Carvalho compounds, under the conditions used in this study. This could help to maximally assess the relevance of the new compounds in this paper based on potency.

We thank the reviewer for these suggestions. We agree with the reviewer that an inappropriate pre-incubation may 'inflate' IC₅₀ potencies for irreversible inhibitors, which is why we now additionally report the requested rate constants for irreversible inactivation for all synthesized inhibitor compounds (initially, we only provided the respective rate constants for our most potent compound **16**; Fig. 4c), as requested. They can now be found in Supplementary Table 1 (that is linked in the main text), the corresponding graphs underlying the kinetic analysis are shown in Supplementary Fig. 8. Importantly, the obtained kinetic data show strong correlations with the previously reported trends from the IC₅₀ values and thus support our general conclusion that we have developed potent irreversible DPP inhibitors.

Moreover, again as suggested by the reviewer, we also determined complementary IC₅₀ data for the 4-oxo- β -lactam compound **6** (4O β L-6; Carvalho *et al.*, 2022; reference 27 in our manuscript) as an additional point-of-reference for our IC₅₀ inhibitory data. We added this data to Table 1 and Supplementary Fig. 2. Please note that in reference 27 in our manuscript, a K_i instead of an IC₅₀ was reported along with a different enzyme inhibition workflow which is why the here determined IC₅₀ differs from the published value. Nevertheless, a direct comparison of the IC₅₀ of this internal control compound with the IC₅₀s of the compounds synthesized in this manuscript confirms their potent inhibitory activities.

2) The use of cell viability as a read-out for target engagement seems very tricky. The data in Figure 5c do not unequivocally demonstrate target engagement. In my personal opinion, the lower viability observed with compounds 7, 8 and 16 might well be the consequence of intrinsic cytotoxicity rather than of DPP9 inhibition. After all, the inhibitor concentration used in this experiment (10 μ M) is rather high and off-target effects can reasonably be anticipated at this

concentration. To amend this situation, at least the effect on viability should be shown for the compounds in the same cells, but not in the presence of mitomycin. Also, dose-dependence of the effect should be investigated. Very honestly, I would actually prefer to see a direct measure of target engagement if this would be within the means of the team.

We agree with the reviewer that target engagement confirmation from cell viability measurements requires careful experimental planning and may be difficult to interpret. We have therefore on the one hand carefully revised the data of the cell viability assay and have also added, as a new experiment, a DPP9 proximity ligation-based cell assay to better demonstrate target engagement in cells.

Cell viability assays

We have revised the data and figure on cell viability data as suggested; this data is now presented as Fig. 5d (previously Fig. 5c) in the main text and Supplementary Fig. 11 (previously Supplementary Fig. 8):

Please note that we had already in the first submission reported the requested additional controls in Supplementary Fig. 8, however as an independent experiment (i.e. the data leading to original Fig. 5c and Supplementary Fig. 8 were two independent experiments and were statistically also evaluated as two independent experiments).

As a standard, we however always also measure the requested controls (i.e. treatment with compounds without MMC addition) along with the MMC-treated samples. We have therefore now taken this already available data to report the results on compound-triggered cell viability in presence and absence of MMC in one conjoint experimental setting. As before, the cell viability data in presence of MMC is reported in the main text, this time as Fig. 5d. The corresponding control data, i.e. inhibitor treatment without MMC, is now reported in Supplementary Fig. 11a.

We agree with the reviewer that it is important to investigate whether these compounds display unspecific cytotoxicity which may hamper the detection of the MMC-dependent effect on cell viability. This question is addressed in the Supplementary Fig. 11a. The graph shows the percentage of cell viability of cells treated with 10 μ M of either compound, including 1G244, for 72 hours. All values are normalized to control cells, meaning HeLa wild-type cells mock treated with DMSO. It is important to note that 10 μ M of 1G244 is the standard concentration used in the field. Importantly, the data show that treatment with 1G244 indeed leads to reduced cell viability, as do also compounds **7** and **8**. This decline is higher than that observed for the corresponding mock (DMSO) treated DPP9^{KO} cells. This observed reduced cell viability may

be due to off-target effects. Importantly, under these conditions, compound **16** stands out, as it is not toxic to cells.

In Supplementary Fig. 11b, we show the percentage of cell viability in the presence of MMC of cells treated with the different compounds. To compare whether the inhibitor leads to a hyper response to MMC, we normalized to the viability in the absence of MMC (compound **X** + MMC normalized to compound **X** - MMC). This allows us to calculate the relative cell viability of the [+MMC +inhibitor] treatment. We then tested whether the inhibitor leads to more sensitivity compared to HeLa wild-type cells. For DPP9^{KO} cells we also calculated the relative cell viability [+MMC]. This analysis verifies previous published data showing that DPP9^{KO} cells are hyper-sensitive to MMC. Similarly, also treatment with 1G244 increases the sensitivity of cells to MMC. Importantly, also compound **16**, which is in itself not-toxic, leads to a significant increase in the sensitivity of cells to genotoxic stress, similar to the effect of 1G244, and to that seen for the DPP9^{KO} cells. In Fig. 5d, the data is presented differently, here each time point is compared to HeLa wild-type cells without MMC treatment.

The corresponding figures (Fig. 5d) and Supplementary Fig. 11 therefore now look as:

Figure 5d. **Inhibition of DPP8/9 increases cellular sensitivity to genotoxic stress.** HeLa wild-type cells were treated with 1 μM MMC and 10 μM of the respective inhibitors. Cell viability was measured after 72 h. DPP9^{KO} cells and 1G244 (10 μM) were analyzed as positive controls, whereas Sitagliptin was used as a negative control. Dashed lines indicate the viability of the controls. The graph shows the mean and error bars indicate the SEM of all individual measurements ($n = 18$ (HeLa wild-type controls), 17 (HeLa DPP9^{KO} control) or 9 (inhibitor treated samples) independent replicates). Data were analyzed by a mixed effect analysis ($*p \leq 0.05$, $**p \leq 0.01$, $***p \leq 0.001$, $****p \leq 0.0001$).

Supplementary Figure 11. **Effect of *N*-phosphono-(*S*)-3-aminopiperidinones on cell viability of HeLa cells.** (a) HeLa wild-type cells were incubated with 10 μ M of the respective inhibitors for 72 hours. DPP9^{KO} cells, and 1G244 (10 μ M) were analyzed as positive controls, whereas the DPP4-selective inhibitor Sitagliptin (10 μ M) was used as a negative control. The graph shows the mean and error bars indicate the SEM of all individual measurements ($n = 18$ (HeLa wild-type and DPP9^{KO} control) or 9 (inhibitor treated samples) independent replicates). Data were analyzed by a mixed effect analysis (* $p \leq 0.05$, ** $p \leq 0.01$, *** $p \leq 0.001$, **** $p \leq 0.0001$). (b) HeLa wild-type cells were treated with 1 μ M MMC and 10 μ M of the respective inhibitors. Cell viability was measured after 72 h. DPP9^{KO} cells and 1G244 (10 μ M) were analyzed as positive controls, whereas Sitagliptin was used as a negative control. Dashed lines indicate the viability of the HeLa wild-type control. Data were normalized for each condition [+MMC]

+inhibitor] separately to the corresponding [-MMC + inhibitor] control. The graph shows the mean and error bars indicate the SEM of all individual measurements ($n = 18$ (HeLa wild-type controls), 17 (HeLa DPP9^{KO} control) or 9 (inhibitor treated samples) independent replicates). Data were analyzed by a mixed effect analysis ($*p \leq 0.05$, $**p \leq 0.01$, $***p \leq 0.001$, $****p \leq 0.0001$).

Please also note that we have also changed the statistical analysis of the data of Fig. 5d and Supplementary Fig. 11, from a two-way ANOVA with Tukey test to a mixed effect analysis to reflect that we used different size groups in the analysis.

We thank the reviewer for his/her comment, since we believe that with these analyses, our experimental setting consisting of a DPP9^{KO} control, 1G244 (10 μ M), the use of the DPP4 inhibitor Sitagliptin (10 μ M) as a negative control, and our compounds in combination now with or without MMC treatment serves as further support (together with the below reported PLA assays) for our claim that we also inhibit DPP9 in cells.

Proximity ligation assay for measuring direct target engagement in cells

To further investigate direct target engagement, we also performed proximity ligation assays (PLA). DPP9 is involved in the repair of DNA double-strand breaks and cleaves BRCA2 to regulate the formation of RAD51 foci. The addition of genotoxic agents such as MMC, which induces DNA double-strand breaks, leads to an interaction of DPP9 and BRCA2, which can be visualized by PLA. Since BRCA2 is a substrate, the addition of DPP9 inhibitors that bind to the active site specifically disrupts this interaction as reported previously (Bolgi et al., 2022; reference 24 in our manuscript). Accordingly, we treated cells with 10 μ M of inhibitors **8** and **16** and observed, as expected for in cell DPP9 target engagement, a significant reduction of BRCA2-DPP9 PLA events. The observed reduced PLA events are comparable to the effects from 10 μ M 1G244 treatment, that also binds to the active site of DPP9, thereby blocking BRCA2 binding to the active site of DPP9. The corresponding results are shown in new Fig. 5c and new Supplementary Fig. 10 and look as:

Figure 5c. **Addition of *N*-phosphono-(*S*)-3-aminopiperidine-2-ones block the interaction between endogenous DPP9 and BRCA2.** Quantification of PLAs showing MMC-induced DPP9-BRCA2 PLA events in HeLa wild-type cells, in the presence of 10 μM of the respective inhibitory compounds ($n = 3$ independent biological replicates, marked by varying shapes). Cells were treated with 300 nM MMC for 24 h and 10 μM of the indicated inhibitors for 1 h prior to fixation. Each dot represents the number of PLA events in a single cell. Data were analyzed by unpaired *t*-test comparisons (**** $p \leq 0.0001$).

Supplementary Figure 10. **Independent biological replicates of PLA quantification, showing MMC-induced DPP9-BRCA2 PLA events in HeLa wild-type cells.** Cells were treated with 300 nM MMC for 24 h and received 10 μM of the indicated inhibitor 1 h prior to fixation. (a) Quantification of PLA events for each replicate ($n = 3$ independent biological replicates, marked by varying shapes). Each dot represents the number of PLA events in a single cell. Data were analyzed by unpaired t -test comparisons ($*p \leq 0.05$, $****p \leq 0.0001$). (b) Representative PLA images showing close proximity between endogenous DPP9 and endogenous BRCA2 in HeLa wild-type cells (+Ctrl.). Inhibitor treatment reduced the PLA events (white dots) compared to the control.

Finally, it is somewhat disturbing that the concentration of reference 1G244 in these studies, is not mentioned.

We have used 10 μM 1G244, which is the concentration commonly used in the field. This concentration is identical to that of the other compounds, including Sitagliptin. We apologize that we only noted the applied concentration of 1G244 in the figure, not the figure legend. We have changed this now and for more clarity, the concentrations of all compounds shown in the figure are now also stated in the figure legend.

Reviewer #2 (Remarks to the Author):

The work reported by the authors is of great interest to the chemical biology and drug discovery communities. Their demonstration of the covalent nature of sulphostin towards DPP opens new horizons, which the authors have already started to investigate with a convincing series of N-phosphonopiperidones.

We thank the reviewer for his/her overall positive feedback to our study.

To demonstrate selectivity beyond DPP4/8/9, the authors have utilised chemoprotomics. I have been asked by the editor to specifically review this part of the manuscript.

Two experiments are being reported:

- 1) one studying the target space of Sulphostin using a general fluorophosphonate-alkyne serine hydrolase probe (results in figure 3) and
- 2) one studying the target space of cpd 11 using tailored probe 9

The experimental part indicates how the raw data were processed before being plotted. The supplementary tables give the list of identified proteins together with the related significance and $\log_2(\text{ratios})$. A table with the initial LFQ values is not, unless I am mistaken, provided. However, because data have been imputed, it would be reasonable to be able to judge how many values for the serine hydrolases have been imputed and how successful is the enrichment (target intensities vs non-target intensities for instance).

We apologize that we have not provided initial LFQ values for the MS data and therefore did not indicate how many values were imputed from the normal distribution. We therefore updated the Supplementary Data 1 and 2 and now provide all initial (\log_2 -transformed) LFQ values additionally to the results of the Student's *t*-test. Please note that "NaN" in Supplementary Data 1 and 2 indicates that the respective protein group was not identified in the respective sample and the LFQ-intensity was imputed from the normal distribution (width 0.3, down shift 1.8). To help the reviewer to get an overview on imputed LFQ-values on serine hydrolases in our data, please find below a corresponding list:

Table 1: Number of imputed LFQ-values of serine hydrolases for statistical analysis in respective categorical groups

Gene name	DMSO control	Labeled with FP and pre-incubated with Sulphostin	Labeled with FP
LYPA1	1	0	0
FASN	0	0	0
PPT1	1	1	1
APEH	1	0	0
LONP1	1	1	0
ACOT7	2	1	0
LYPA2	1	0	0

BCHE	4	1	0
ESD	1	0	0
TPP2	0	0	0
PREP	0	0	0
PAFAH1B2	2	0	0
PAFAH1B3	1	0	0
PREPL	2	0	0
LYPLAL1	1	0	1
DPP8	4	4	0
DPP9	1	1	0
ACOT1, ACOT2	2	0	0
ABHD11	3	0	0
CNDP2	2	1	4
PAFAH2	4	2	0
ABHD6	4	1	0
CPVL	2	0	0
ABHD10	2	0	0
PPME1	1	0	0

In this line of thought, it would be important for the reader to be informed about the success of the enrichment with the fluorophosphonate-alkyne: how many serine hydrolases are usually captured by other groups using this probe and workflow?

We would like to thank the reviewer for his/her helpful comment. In order to analyze the number of serine hydrolases identified in the ABPP experiment, we screened our dataset against a published list enumerating all human serine hydrolases (Bachovchin & Cravatt, 2012). According to this list, we identified 25 serine hydrolases in our experiment, of which 19 serine hydrolases were significantly enriched ($p \leq 0.01$). Of note, only 21 proteins were significantly enriched by the application of FP-alkyne demonstrating the high selectivity of the probe and our workflow. Of all 237 serine hydrolases listed in the corresponding publication, according to the Human Proteome Atlas, 156 serine hydrolases are expressed in HEK cells, however several of them only at low levels or condition-dependent, making it difficult to estimate the “success rate” of identified serine hydrolases in the sample. Please note that a direct comparison of our identification rate to experiments of other groups is challenging as these studies have often been performed with different probes (e.g. biotin-FP instead of alkyne-FP that requires one further click step), cell lines and workflows (e.g. LFQ-based quantification vs. SILAC, etc.).

We nevertheless extracted some analyses in human cells from the literature. For example, ~30 serine hydrolases were identified by the application of 5-10 μ M FP-biotin in human prefrontal cortex total homogenates (Cisar *et al.*, 2018; now cited as ref. 41 in our manuscript) and in PC3 cell proteomes (Chang, Nomura, & Cravatt, 2011). More recently, Kok and colleagues identified 49 and 55 serine hydrolases by the application of 5 μ M FP-alkyne in HepG2 cells lysates (Kok *et al.*, 2020; now cited as ref. 40 in our manuscript). Finally, in our

previous study on β -oxo-lactam-based DPP8/9 inhibitors, we used 10 μ M FP-biotin and SILAC quantification in a competitive ABPP approach and enriched 49 serine hydrolases (the corresponding experiments were performed in the Cravatt lab, Carvalho *et al.*, 2022; ref. 27 in our manuscript).

We therefore have lower identification rates, most probably as a result of a lower probe concentration in our assays (which we applied to lower non-serine hydrolase background labeling by FP-alkyne), an additional click-step and LFQ-based quantification. We have added a corresponding comment to this finding to the main text.

We however would like to emphasize that these competitive ABPP experiments only served to demonstrate that the Sulphostin scaffold displays a promising target selectivity for DPP-like serine proteases, making it an interesting starting structure for inhibitor design (the corresponding statement in the MS is: "...revealing Sulphostin as a promising starting structure for further inhibitor development.") We do not claim that Sulphostin is an exclusive DPP8/9 inhibitor or that we have demonstrated this. We therefore do not believe that further experimental efforts to improve identification rates would significantly improve the quality of the manuscript.

Continuing on the first experiment (figure 3), the manuscript indicates that 21 protein groups, mostly serine hydrolases were significantly enriched by the general probe. I believe that all serine hydrolases measured in this experiment should be annotated with another colour on the volcano plot. The protein within the threshold decided by the authors contains 6 proteins where only 2 are annotated: DPP8 and DPP9. It is not clear at first sight that the red square are the only ones that pass a significance criterium, the unannotated red square being therefore PYCR2. This figure would benefit from a legend on the figure, and the authors could consider plotting the size of the dots according to significance rather than a hard arbitrary threshold. Any reader will wonder what is the significance of the 4 remaining proteins in the "target quadrant". Additionally, instead of "competitor" and "probe" their identity should feature in the axis labels.

We thank the reviewer for his/her suggestion. As described above, we identified 25 serine hydrolases in our competitive ABPP dataset with FP-alkyne as probe and Sulphostin as competitor, according to the published list of all human serine hydrolases (Bachovchin & Cravatt, 2012). To increase the requested transparency in the presentation of our data set, we marked all identified serine hydrolases with a red symbol in the respective figure (Fig. 3b). We however used different symbols (circles, triangles, etc.) to indicate the corresponding significance level in both the enrichment as well as in the competitive ABPP experiment. For both, we chose $p \leq 0.01$ as the significance threshold. Moreover, we changed the axis label

and labelled the figure with the probe name (FP-alkyne) and competitor name (Sulphostin), as requested.

More importantly, I believe that the claim of selectivity would benefit from an amendment of the experiment seen in 5b. As such, the following elements do not allow for me to be perfectly convinced:

a) In the first experiment with a general probe (figure 3b) DPP8 can be enriched and competed by sulphostin despite a mediocre IC50. However in the final experiment DPP8 cannot be seen in the expected quadrant despite the better potency of 9 and 11 against DPP8 vs DPP9, the latter being expectedly found. Since this target is missed, it is quite difficult to be convinced that other targets are not also missing from the profile.

B) Compounds 5, 7, 8 & 16 are being tested in cells, not 11. The selectivity of 11 is thereby of little interest, while the authors put forward 8 and 16 as their favourite molecules.

The chemoproteomics experiment should thereby concern molecules 8 and/or 16 as competitors with 9 as the enrichment probe.

We thank the reviewer for his/her comment and would like to answer both questions together. We agree that the absence of DPP8 in the expected quadrant may affect the conviction of the full assessment of the target spectrum of our compounds and that chemoproteomics experiments with additional inhibitors will be helpful to better characterize the target selectivities of the *N*-phosphonopiperidones. We however want to make the reviewer aware that reliable DPP8 detection in standard cell lines is difficult due to its low expression levels (in relation to DPP9 but also other serine hydrolases) which is why determination of DPP8 competitive ABPP inhibition values is usually assessed in overexpression settings (as for example performed in the 4-oxo- β -lactam covalent inhibitor study (Carvalho *et al.*, 2022; reference 27 in our manuscript) which however does not allow to deduce selectivities. We also originally decided to perform the chemoproteomic analysis with compound **11**, as it is the non-tagged structural analogue to compound **9**.

To address these concerns and as suggested by the reviewer, we therefore performed an additional set of chemoproteomics experiments and now also analyzed compounds **8** and **16** together with the previously analyzed compound **11** in one large competitive ABPP experiment. The corresponding data is now presented as new Fig. 5b and Supplementary Data 2 and basically confirms that also these compounds specifically inhibit labeling of DPP9. In addition, compounds **8** and **16** display a better selectivity vs. PREP than the previously investigated compound **11** which is an interesting finding.

Unfortunately however, DPP8 again does not appear in the “significance” quadrant (significantly enriched with **9** and significantly competed with **8**, **11**, or **16**). However, we believe that this is a consequence of data imputation and the overall low levels of DPP8 in cells. Indeed, a more detailed analysis of the raw data shows that DPP8 was only identified in the samples treated with the alkyne-tagged probe **9** (in 3 out of 4 replicates), whereas no peptides of DPP8 were identified in the DMSO control and in the samples pre-incubated with compounds **8**, **11**, and **16**. This shows that DPP8 was enriched by the application of the probe **9** and that previous application of the corresponding inhibitors prevents DPP8 enrichment (resulting in missing values “NaN”). However, to enable statistical data processing, the “NaN” values are then “filled up” from the imputation of the missing values from the normal distribution, which, due to the low levels of DPP8 in cells, leads to imputed LFQ levels that match endogenous DPP8 levels (please note that we have used settings that are typically used in this field and are preset by the program, i.e. width 0.3, downshift 1.8). Importantly, DPP9 is significantly more abundant (~10-fold) which explains the differences in the initial values and the corresponding log₂-fold changes (Geiss-Friedlander *et al.*, 2009). Please find below a summary of the different LFQ results for DPP8 under different conditions, illustrating the problems resulting from the imputation procedure (which shows that the imputed LFQ intensities are unfortunately in the same range as the LFQ intensities for treatment with probe **9**, thereby hampering statistical analysis of significance).

Table 2: Initial and imputed (log₂-transformed) LFQ intensities of DPP8.

	#sample no.	Initial LFQ intensity	Imputed LFQ intensity
DMSO control	1	NaN	19.895
	2	NaN	19.1469
	3	NaN	19.1378
	4	NaN	20.0301
Labeled with 9 , competed with 8	5	NaN	18.7579
	6	NaN	19.4636
	7	NaN	19.9898
	8	NaN	20.0284
Labeled with 9 , competed with 11	9	NaN	20.0178
	10	NaN	19.0548
	11	NaN	20.1302
	12	NaN	20.0267
Labeled with 9 , competed with 16	13	NaN	19.0933
	14	NaN	18.7779
	15	NaN	19.1967
	16	NaN	19.1641
Labeled with 9	17	21.9345	21.9345
	18	NaN	19.5622
	19	21.2153	21.2153
	20	19.9933	19.9933

We are aware that this imputation problem could be avoided by using different settings for imputation (e.g. by using a larger downshift). However, from our point of view, this would manipulate the results more strongly, which is why we have refrained from doing this for better transparency and have analyzed the data with the “classical” settings.

Additionally, as mentioned for figure 3b, the legend of figure 5b could be made more explicit for the sake of clarity.

We included a legend in Fig. 5b comparable to Fig. 3b.

Compared to figure 3a, also, only the warhead remains (more similarly to ligand-directed chemistry) on the target proteins, rendering the scheme in figure 3a somewhat inaccurate to explain the latter experiment.

We agree with the reviewer that our presentation of the workflow does not optimally display the “ligand-directed”-type labeling mediated by the Sulphostin analogues. We have therefore changed the corresponding representation of the “inhibitor” in the figure to make this unusual mode-of-action clearer. However, we want to keep the presentation for the “ABP probe” as we both used FP-alkyne (a classical ABP) and the *N*-phosphonopiperidone-based ABP (a “ligand-directed”-type probe) in our chemoproteomics workflow which is otherwise difficult to visualize in one conjoint workflow presentation.

Other Minor comments:

1) A few Germanisms have slipped through the authors attention and should be attended to:

a) a few "phosphor" instead of "phosphorus" in p6

Changed accordingly.

b) "sulfon" without a "e" p9

Changed accordingly.

c) "applied" l294

Changed accordingly.

2) The number of significant digits for many values reported in the manuscript are not reasonably representing the estimated experimental errors.

We carefully went through the whole manuscript and amended the significant digits where appropriate.

Reviewer #3 (Remarks to the Author):

The main discoveries of the manuscript "Sulphostin-inspired N-phosphonopiperidones as selective covalent DPP8 and DPP9 inhibitors" are (1) Sulphostin was shown to be a covalent inhibitor of DPP4/9, (2) high-resolution crystallography confirmed the binding mode of Sulphostin, (3) proteomic analysis showed that only DPP8/9 binds to Sulphostin intracellularly, (4) various Sulphostin derivatives were synthesized and compound 16 was found, (5) the complex structure of compound 16 with DPP9 was determined and the compound 16 was shown to be a covalent inhibitor, (6) intracellular targeting tests showed that compounds 9 and 11 bind to DPP9 and PREP, and (7) investigations of the effect of DPP8/9 inhibitors on cell sensitivity to genotoxic agents showed that the tested compounds were promising target selective DPP8/9 inhibitors, which are active in cells. Thus, this research was conducted in a very logical manner, utilizing methods from diverse fields such as biochemistry, physical chemistry, organic chemistry, and cell biology.

As mentioned above, this study has yielded a number of useful data, but the most interesting point is that target selectivity is ensured by the leaving group of the inhibitor. Not only for DPP9, but many covalent inhibitors for serine proteases mimic acyl enzyme intermediates. In such cases, the specificity is ensured by the N-terminal fragment structure that binds to the catalytic Ser, and the structure of the leaving group (the C-terminal fragment of the cleaved peptide, analogous to a peptide structure) has not received much attention.

I appreciate the extensive work carried out by authors which may be useful in developing novel DPP9 inhibitors. I enjoyed reading the manuscript. However, I have some comments that I believe will enhance the clarity and impact of the author's findings.

We thank the reviewer for his/her overall positive feedback to our manuscript.

MAJOR comments

1. Comparing the structure of Sulphostin to the substrate peptide structure, the leaving group of Sulphostin (3-aminopiperidin-2-one moiety) corresponds to the P2-P1 residue of a substrate peptide, and a phosphosulfamate moiety to P1'-P2'. On the other hand, in peptide cleavage by DPP9, P1'-P2' is removed to form an acyl intermediate, P2-P1-OG(Ser730). This difference in reaction mechanism is very interesting from an enzymatic chemistry point of view and should definitely be discussed in more detail. Specifically, please add a figure of the reaction

mechanism of acyl intermediate formation in peptide cleavage by DPP9 to Fig. 2 (as Fig. 2f) or provide an additional supplementary figure showing a comparison with Fig. 2e.

We thank the reviewer for his/her interesting suggestion. We included a corresponding scheme as a supplementary figure (Supplementary Fig. 4) and discussed the difference in the reaction mechanism more in detail in our manuscript.

2. Have you performed any experiments using the 3R epimer of Sulphostin in the evaluation of DPP9 inhibitory activity or in the structural analysis of its complex with DPP9?

We thank the reviewer for his/her comment on the epimers of Sulphostin. We now synthesized all stereochemical combinations of Sulphostin and tested all of them in our biochemical assay:

	Configuration (C-3,P)	IC ₅₀ [nM]		
		DPP4	DPP8	DPP9
WTC0260A	S,S	1,438 ± 58	>100,000	72,714 ± 63,746
WTC0260B	R,R	21 ± 1	27,904 ± 19,148	6,261 ± 928
WTC0260C	R,S	5,411 ± 427	>100,000	>100,000
Sulphostin	S,R	79 ± 29	6,930 ± 620	1,392 ± 108

We found that the P-isomers have generally a lower inhibitory potential, while the 3R epimer of Sulphostin is active towards all DPP proteins. Indeed, this analogue is even more potent against DPP4 than the parent natural product, whereas DPP8 and 9 are less potently but still efficiently inhibited. These findings are thus in agreement with literature reports (Abe *et al.*, 2004; ref. 37 in our study).

3. Complexes of DPP9 and DPP4 with Sulphostin were obtained by the soaking method, which is reasonable evidence in that Sulphostin is a covalent inhibitor. However, there is a concern that the authors may have failed to detect the structural changes caused by Sulphostin binding due to the limitations by crystal packing. On the other hand, the complex of compound 16 with DPP9 was obtained by co-crystallization, so there is little risk of missing the structural changes caused by inhibitor binding. Therefore, by comparing the structures of the DPP9/compound

16 complex and the DPP9/Sulphostin complex, the authors may be able to check the above concerns in the DPP9/Sulphostin complex structure.

We thank the reviewer for raising this concern. While the soaking method is sometimes limited by crystal packing, we optimized the conditions for Sulphostin binding to ensure effective interaction with DPP9 and DPP4, and we observed clear electron density for Sulphostin in both complexes. While crystal packing could potentially obscure subtle structural changes, we did not detect major conformational differences in the overall structure. We agree that comparing the DPP9/Sulphostin complex with the DPP9/compound **16** complex (obtained by co-crystallization) is a valuable approach to further assess any structural differences caused by inhibitor binding. The overlay of the corresponding structures clearly shows that the amino acid residues of the active site, such as E248 and N810, make a conformational change after ligand binding, while the backbone is almost congruent.

Figure 1: Overlay of X-ray structures DPP9:1 (shown in blue) and DPP9:16 (shown in black). Active site residues are shown as stick model, whereas the backbone of DPP9 is shown as ribbon model.

Interestingly, however, a significant difference can be seen in the crystal structures, as the R-helix could be resolved in the structure of Sulphostin:DPP9, while the R-helix could not be resolved in the apo structure and in DPP9:16, probably due to a high flexibility. However, it is known that the helix becomes ordered upon ligand binding, leading to electron density in the crystal structure (Ross *et al.*, 2018; reference 17 in our manuscript). We do not believe that this is a consequence of the crystallization methods, but rather caused by steric hindrance of our ligand with certain residues of the R-helix. In detail, compound **16** most likely collides with R133, which can explain a disordered R-segment.

MINOR comments

1. lines 568-569, 574-575 and 580-581, please unify the notations: TRIS-HCl or tris, pH x or pH=x.xx

- Changed accordingly.

2. lines 569 and 575, 0.10 Tris -> 0.10 M Tris

- Changed accordingly.

3. line 582, a 1:1 ration at 293 K -> a 1:1 ratio at 293 K.

- Changed accordingly.

4. In Fig. 1a, the hydrogen bond between the double Glu motif (E248 and E249) and the water molecule occupying the position corresponding to the 3-position nitrogen of the Sulfostin (corresponding to the N-terminal of the substrate peptide) should also be shown.

We thank the reviewer for his/her comment. We checked the respective crystal structures from PDB (1PFQ, 6EEO and 6EOQ) that were used for these figures. However, in these published structures, no water molecule has been resolved at the corresponding position of the 3-position nitrogen of Sulphostin.

5. In Tables 1 and 2, the IC₅₀ value of 1G244 for DPP4 seems to be evaluated only at relatively low concentrations (indicated as >100 nM) as compared with other compounds. What happens when tested at higher concentrations? Is it not possible to test at higher concentrations for 1G244 due to solubility issues, etc.?

We thank the reviewer for this suggestion. Indeed, we did not test higher concentrations of 1G244, as previous studies on this inhibitor already showed that its IC₅₀ value for DPP4 is above 100 μM (Jiaang *et al.*, 2005; Wu *et al.*, 2009; references 32 and 38 in our manuscript). However, in order to validate this finding in our experimental set up, we performed inhibition assays with DPP4 at concentrations up to 100 μM. Indeed, although we observed moderate inhibition at the highest concentration (100 μM, corresponding to 60% activity loss), there was no significant inhibition in the tested concentration range. Moreover, this data was also added to Table 1 as well as Supplementary Fig. 2.

6. Table S2 (Data Collection), What criteria did you use to determine the upper limit of resolution for the diffraction data? According to Table S2, the CC_{half} value of the outer shell is well above 0.5 for each of the data, so I would think that a little higher resolution data would be available.

We thank for suggesting the possibility of incorporating additional data into the processing step based on the cc1/2 criterion, which could enhance the resolution. Data processing was carried out using AutoProc from Global Phasing, and we adhered to the default resolution cutoff with Mean(I)/σ > 1.2.

7. Table S2 (Refinement), The size of the test set used for refinement is quite small, less than 1%, what is your aim? Please comment.

We appreciate the reviewer's careful examination of the data. For the DPP9 structures, with more than 40,000 processed reflections, over 2,000 reflections were set aside for the test set. It is generally accepted that excluding more than 2,000 reflections from refinement yields minimal benefit, which is why this results in a relatively low percentage of reflections in the test set. Regarding the DPP4 structure, while the fraction and total number of reflections in the test set are at the lower end, they remain statistically meaningful, as the number is still above 1,000.

References

- Abe, M. *et al.* First synthesis and determination of the absolute configuration of sulphostin, a novel inhibitor of dipeptidyl peptidase IV. *J Nat Prod* **67**, 999-1004 (2004).
- Bachovchin, D. A. & Cravatt, B. F. The pharmacological landscape and therapeutic potential of serine hydrolases. *Nat Rev Drug Discov* **11**, 52-68 (2012).
- Bolgi, O. *et al.* Dipeptidyl peptidase 9 triggers BRCA2 degradation and promotes DNA damage repair. *Embo Reports* **23** (2022).
- Carvalho, L. A. R. *et al.* Chemoproteomics-Enabled Identification of 4-Oxo-beta-Lactams as Inhibitors of Dipeptidyl Peptidases 8 and 9. *Angew Chem Int Ed Engl* **61**, e202210498 (2022).
- Chang, J. W., Nomura, D. K. & Cravatt, B. F. A Potent and Selective Inhibitor of KIAA1363/AADACL1 that Impairs Prostate Cancer Pathogenesis. *Chemistry & Biology* **18**, 476-484 (2011).
- Cisar, J. S. *et al.* Identification of ABX-1431, a Selective Inhibitor of Monoacylglycerol Lipase and Clinical Candidate for Treatment of Neurological Disorders. *Journal of Medicinal Chemistry* **61**, 9062-9084 (2018).
- Geiss-Friedlander, R. *et al.* The Cytoplasmic Peptidase DPP9 Is Rate-limiting for Degradation of Proline-containing Peptides. *Journal of Biological Chemistry* **284**, 27211-27219 (2009).
- Jiaang, W. T. *et al.* Novel isoindoline compounds for potent and selective inhibition of prolyl dipeptidase DPP8. *Bioorg Med Chem Lett* **15**, 687-691 (2005).
- Kok, B. P. *et al.* Discovery of small-molecule enzyme activators by activity-based protein profiling. *Nat Chem Biol* **16**, 997-1005 (2020).
- Ross, B. *et al.* Structures and mechanism of dipeptidyl peptidases 8 and 9, important players in cellular homeostasis and cancer. *Proc Natl Acad Sci U S A* **115**, E1437-E1445 (2018).
- Wu, J. J. *et al.* Biochemistry, pharmacokinetics, and toxicology of a potent and selective DPP8/9 inhibitor. *Biochem Pharmacol* **78**, 203-210 (2009).

REVIEWERS' COMMENTS

Reviewer #1 (Remarks to the Author):

I appreciate the authors' thoughtful replies to my remarks and I also agree with the additional experimental work that they have done and the clarifications they have made in the manuscript. Overall, I feel that my remarks have been adequately addressed and I therefore have no objections that this manuscript is published in Nature Communications.

We thank the reviewer for his/her positive feedback and his/her recommendation for publishing our study in its present form.

Reviewer #2 (Remarks to the Author):

I would like to thank the authors for the additional experiments and amendments they made to their manuscript. I am glad to see that my request led to find that the best molecules do not hit PREP. I believe the manuscript is suitable for publication and I wish great impact to the authors.

I would like to add however the following, mainly as food for thoughts: I think that the choice of imputing values this way for this type of experiments does not do justice to the inhibitors. This type of imputation that assumes a normal distribution of the proteins close to the level of detection is valid for full proteomes; is it really for enrichment experiments? I am convinced that DPP8 is bound by the 3 molecules and that 0 values compared to 3 values in the "competition" should be good enough to call it a target. However, I do appreciate the dictatorship of the p-value in the field that leads to use imputation even if doubtful.

We thank the reviewer for his/her positive feedback and his/her recommendation for publishing our study in its present form.

We agree with the reviewer that the imputation of missing values from the normal distribution in the corresponding experiment is not a perfect solution for an enrichment experiment. We are convinced, in agreement with the reviewer, that DPP8 is also bound by our three compounds, which was also confirmed by the corresponding enzyme assays. Nevertheless, the analysis of the LFQ intensities requires the "existence of values" which is why missing values have to be imputed to obtain a p-value. As the reviewer rightly notes, this is a common method and expectation in the field which is why we have also made this type of analysis, although we are aware of its inherent limitations.

Reviewer #3 (Remarks to the Author):

I have read the revised version of Sewald et al. and confirm that my comments on the first version have been adequately addressed in the revised version.

I therefore consider the revised version to be substantially improved and worthy of publication in Nature Communications.

We thank the reviewer for his/her positive feedback and his/her recommendation for publishing our study in its present form.